# RETHINK DEPTH SEPARATION WITH INTRA-LAYER LINKS

## ABSTRACT

The depth separation theory is nowadays widely accepted as an effective explanation for the power of depth, which consists of two parts: i) there exists a function representable by a deep network; ii) such a function cannot be represented by a shallow network whose width is lower than a threshold. Here, we report that adding intra-layer links can greatly improve a network's representation capability through the bound estimation, explicit construction, and functional space analysis. Then, we modify the depth separation theory by showing that a shallow network with intra-layer links does not need to go as wide as before to express some hard functions constructed by a deep network. Such functions include the renowned "sawtooth" functions. Our results supplement the existing depth separation theory by examining its limit in a broader domain. Also, our results suggest that once configured with an appropriate structure, a shallow and wide network may have expressive power on a par with a deep network.

## 1 INTRODUCTION

Due to the widespread applications of deep networks in many important fields (LeCun et al., 2015), mathematically understanding the power of deep networks has been a central problem in deep learning theory (Poggio et al., 2020). The key issue is figuring out how expressive a deep network is or how increasing depth promotes the expressivity of a neural network better than increasing width. In this regard, there have been a plethora of studies on the expressivity of deep networks, which are collectively referred to as the depth separation theory.

A popular idea to demonstrate the expressivity of depth is the complexity characterization that introduces appropriate complexity measures for functions represented by neural networks (Pascanu et al., 2013; Montufar et al., 2014; Telgarsky, 2015; Montúfar, 2017; Serra et al., 2018; Hu & Zhang, 2018; Xiong et al., 2020; Bianchini & Scarselli, 2014; Raghu et al., 2017), and then reports that increasing depth can greatly boost such a complexity measure. In contrast, a more concrete way to show the power of depth is to construct functions that can be expressed by a small network of a given depth, but cannot be approximated by shallower networks, unless its width is sufficiently large (Telgarsky, 2015; 2016; Arora et al., 2016; Eldan & Shamir, 2016; Safran & Shamir, 2017; Venturi et al., 2021). For example, Eldan & Shamir (2016) constructed a radial function and used Fourier spectrum analysis to show that a two-hidden-layer network can represent it with a polynomial number of neurons, but a one-hidden-layer network needs an exponential number of neurons to achieve the same level of error. Telgarsky (2015) employed a ReLU network to build a one-dimensional "sawtooth" function whose number of pieces scales exponentially over the depth. As such, a deep network can construct a sawtooth function with many pieces, while a shallow network cannot unless it is very wide. Arora et al. (2016) derived the upper bound of the maximal number of pieces for a univariate ReLU network, and used this bound to elaborate the separation between a deep and a shallow network. In a broad sense, we summarize the elements of establishing a depth separation theorem as the following: i) there exists a function representable by a deep network; ii) such a function cannot be represented by a shallow network whose width is lower than a threshold.

The depth separation theory is nowadays widely accepted as an effective explanation for the power of depth. However, we argue that depth separation does not hold when we slightly adjust the structure of the shallow networks. Our investigation is on ReLU networks. As shown in Figure 1(c), inspired by ResNet that adds residual connections across layers, we add residual connections within

a layer, which forces a neuron to take the outputs of its neighboring neuron. Then, we find that inserting intra-layer links can greatly increase the maximum number of pieces represented by a shallow network. As such, we can modify the statement of depth separation: without the need of going as wide as before, a shallow network can express as a complicated function as a deep network could.

Our result is valuable in two aspects. On the one hand, it non-trivially supplements the depth separation theory. In reality, a neural network often is not feedforward but uses shortcuts to link distant layers to facilitate feature reuse and easy training. Exploring the depth separation in the shortcut paradigm reveals the limit of the existing theory and motivates us to rethink the genuine power of depth. On the other hand, the superiority of depth over width nowadays seems to become a doctrine for most deep learning practitioners. However, studies such as width-depth equivalence (Fan et al., 2020) and representation ability of wide networks (Lu et al., 2017; Levine et al., 2020) show the essential role of width. In the same vein, our work suggests the potential of wide networks in expressivity. Once configured with an appropriate structure, a shallow and wide network may have expressive power on a par with a deep network. Note that adding intra-layer links is not equivalent to increasing depth. The common understanding of increasing depth is to increase the number of layers, while intra-layer links are just to connect neurons in the same layer, which is a slight change to the original network.

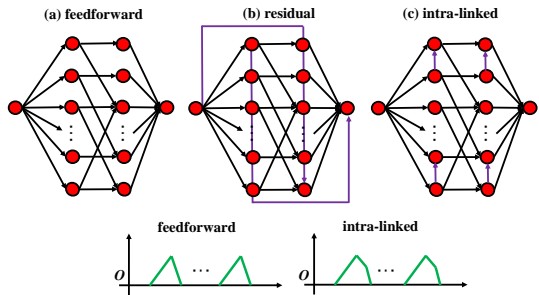

Figure 1: (a) feedforward, (b) residual, and (c) intra-linked.

Specifically, our roadmap to the modification of depth separation theorems includes two milestones. 1) Through bound analysis (Theorems 4, 6, and 8), explicit construction (Propositions 1, 2, and 3), and functional space analysis (Theorem 10), we substantiate that a network with intra-layer links can produce much more pieces than a feedforward network, and the gain is at most exponential, *i.e.*, $\left(\frac{3}{2}\right)^k$, where $k$ is the number of hidden layers. 2) Since intra-layer links can yield more pieces, they can be used to modify depth separation theorems by empowering a shallow network to represent a function constructed by a deep network, even if the width of this shallow network is lower than the prescribed threshold. The modification is done in the cases of $k^2$ vs 3 (Theorem 12) and $k^2$ vs $k$ (Theorem 13, the famous sawtooth function (Telgarsky, 2015)). Also, we point out that the depth separation cannot be fully accomplished based on the bound analysis, unless the bound is proved to be tight. Thus, Arora et al. (2016)'s depth separation theorem might need to be re-examined.

To summarize, our contributions are threefold. 1) We point out the limitation of the depth separation and propose to consider inserting intra-layer links in shallow networks. 2) We show via bound estimation, explicit construction, and functional space analysis that intra-layer links can make a ReLU network produce more pieces. 3) We modify the depth separation result including the famous Telgarsky (2015)'s theorem by demonstrating that a shallow network with intra-layer links does not need to go as wide as before to represent a function constructed by a deep network.

## 2 RELATED WORK

Recently, a plethora of depth separation studies have shown the superiority of deep networks over shallow ones from the points of view of complexity analysis and constructive analysis.

The complexity analysis is to characterize the complexity of the function represented by a neural network, thereby demonstrating that increasing depth can greatly maximize such a complexity measure. Currently, one of the most popular complexity measures is the number of linear regions because it conforms to the functional structure of the widely-used ReLU networks. For example, Pascanu et al. (2013); Montufar et al. (2014); Montúfar (2017); Serra et al. (2018); Hu & Zhang (2018); Hanin & Rolnick (2019) estimated the bound of the number of linear regions generated by a fully-connected ReLU network by applying Zaslavsky's Theorem (Zaslavsky, 1997). Xiong et al. (2020) offered the first upper and lower bounds of the number of linear regions for convolutional networks. Other complexity measures include classification capabilities (Malach & Shalev-Shwartz, 2019), Betti numbers (Bianchini & Scarselli, 2014), trajectory lengths (Raghu et al., 2017), global curvature (Poole et al., 2016), and topological entropy (Bu et al., 2020). Please note that using complexity measures to justify the power of depth demands a tight bound estimation. Otherwise,

it is insufficient to say that shallow networks cannot be as powerful as deep networks, since deep networks cannot reach the upper bound.

The construction analysis is to find a family of functions that are hard to approximate by a shallow network, but can be efficiently approximated by a deep network. Eldan & Shamir (2016) built a special radial function that is expressible by a 3-layer neural network with a polynomial number of neurons, but a 2-layer network can do the same level approximation only with an exponential number of neurons. Later, Safran & Shamir (2017) extended this result to a ball function, which is a more natural separation result. Venturi et al. (2021) generalized the construction of this type to a non-radial function. Telgarsky (2015; 2016) used an $\mathcal{O}(k^2)$-layer network to construct a sawtooth function. Given that such a function has an exponential number of pieces, it cannot be expressed by an $\mathcal{O}(k)$-layer network, unless the width is $\mathcal{O}(\exp(k))$. Arora et al. (2016) estimated the maximal number of pieces a network can produce, and established the size-piece relation to advance the depth separation results from $(k^2, k)$ to $(k, k')$, where $k' < k$. Other smart constructions include polynomials (Rolnick & Tegmark, 2017), functions of a compositional structure (Poggio et al., 2017), Gaussian mixture models (Jalali et al., 2019), and so on. Our work also highlights the construction, and we use an intra-linked network to more efficiently build a sawtooth function.

## 3  NOTATION AND DEFINITION

**Notation 1** (Feedforward networks). For an $\mathbb{R}^{w_0} \to \mathbb{R}$ ReLU DNN with widths $w_1, \ldots, w_k$ of $k$ hidden layers, we use $\mathbf{f}_0 = \left[ f_0^{(1)}, \ldots, f_0^{(w_0)} \right] = \mathbf{x} \in \mathbb{R}^{w_0}$ to denote the input of the network. Let $\mathbf{f}_i = \left[ f_i^{(1)}, \ldots, f_i^{(w_i)} \right] \in \mathbb{R}^{w_i}, i = 1, \cdots, k$, be the vector composed of outputs of all neurons in the $i$-th layer, for $i = 1, \ldots, k$, $j = 1, \ldots, w_i$, we use $g_i^{(j)} = \left\langle \mathbf{a}_i^{(j)}, \mathbf{f}_{i-1} \right\rangle + b_i^{(j)}$ to denote the pre-activation of the $j$-th neuron of the $i$-th layer, where $\mathbf{a}_i^{(j)} \in \mathbb{R}^{w_{i-1}}, b_i^{(j)} \in \mathbb{R}$ are parameters. Then $f_i^{(j)} = \sigma \left( g_i^{(j)} \right)$, where $\sigma(\cdot)$ is the ReLU activation. The output of this network is $g_{k+1} = \langle \mathbf{a}_k, \mathbf{f}_k \rangle + b_k$, where $\mathbf{a}_k \in \mathbb{R}^{w_k}, b_k \in \mathbb{R}$ are parameters.

**Notation 2** (Intra-linked networks). For an $\mathbb{R}^{w_0} \to \mathbb{R}$ ReLU DNN with every 2 paired neurons linked in each hidden layer and widths $w_1, \ldots, w_k$ of $k$ hidden layers, we use $\tilde{\mathbf{f}}_0 = \left[ \tilde{f}_0^{(1)}, \ldots, \tilde{f}_0^{(w_0)} \right] = \mathbf{x} \in \mathbb{R}^{w_0}$ to denote the input of the network. Let $\tilde{\mathbf{f}}_i = \left[ \tilde{f}_i^{(1)}, \ldots, \tilde{f}_i^{(w_i)} \right] \in \mathbb{R}^{w_i}$ be the vector composed of the neurons in the $i$-th layer, then for $i = 1, \ldots, k$, $j = 1, \ldots, w_i$, we use $\tilde{g}_i^{(j)} = \left\langle \tilde{\mathbf{a}}_i^{(j)}, \tilde{\mathbf{f}}_{i-1} \right\rangle + \tilde{b}_i^{(j)}$ to denote the pre-activation of the $j$-th neuron in the $i$-th layer, where $\tilde{\mathbf{a}}_i^{(j)} \in \mathbb{R}^{w_{i-1}}, \tilde{b}_i^{(j)} \in \mathbb{R}$ are some parameters. In an intra-linked network, the $j$-th and $(j+1)$-th neurons are linked, and the $(j+2)$-th and $(j+3)$-th neurons are linked, we prescribe $\tilde{f}_i^{(j)} = \sigma \left( g_i^{(j)} \right)$, $\tilde{f}_i^{(j+1)} = \sigma \left( \tilde{g}_i^{(j+1)} - \tilde{f}_i^{(j)} \right)$, $\tilde{f}_i^{(j+2)} = \sigma \left( \tilde{g}_i^{(j+2)} \right)$, $\tilde{f}_i^{(j+3)} = \sigma \left( \tilde{g}_i^{(j+3)} - \tilde{f}_i^{(j+2)} \right)$. Similar with the classical network, the output of the network is $\tilde{g}_{k+1} = \left\langle \tilde{\mathbf{a}}_k, \tilde{\mathbf{f}}_k \right\rangle + \tilde{b}_k$, where $\tilde{\mathbf{a}}_k \in \mathbb{R}^{w_k}, \tilde{b}_k \in \mathbb{R}$ are parameters.

**Notation 3** (Sawtooth function). A piecewise linear (PWL) function $g : [a, b] \to \mathbb{R}$ is of "$N$-sawtooth" shape, if $g = (-1)^{n-1} \left( x - (n-1) \cdot \frac{b-a}{N} \right), x \in \left[ (n-1) \cdot \frac{b-a}{N}, n \cdot \frac{b-a}{N} \right]$, for $n \in [N]$.

**Definition 1** (Width and depth of a feedforward network (Arora et al., 2016)). *For any number of hidden layers $k \in \mathbb{N}$, input and output dimensions $w_0, w_{k+1} \in \mathbb{N}$, an $\mathbb{R}^{w_0} \to \mathbb{R}^{w_{k+1}}$ feedforward network is given by specifying a sequence of $k$ natural numbers $w_1, w_2, \ldots, w_k$ representing widths of the hidden layers. The depth of the network is defined as $k + 1$. The width of the network is $\max \{ w_1, \ldots, w_k \}$.*

**Definition 2** (Width and depth of a shortcut network (Fan et al., 2020)). *Given a shortcut network $\Pi$, we delete the minimum number of links to make the resultant network $\Pi'$ a feedforward network without isolated neurons. Then, we define the width and depth of $\Pi$ as the width and depth of $\Pi'$.*

Admittedly, defining the width and depth of a network embedded with shortcuts is tricky. A reasonably good definition should conform to the customary understanding of depth and width. For

example, the ResNet (He et al., 2016) shall not be taken as a wide network in the light of the proposed definition; otherwise, it conflicts with practitioners' common sense. Our definition for width and depth can fit the common sense for the network that is formed by slightly modifying a feedforward network, *e.g.*, ResNet, DenseNet (Huang et al., 2017), and S3Net (Fan et al., 2018). Per our definition, the width and depth of intra-linked networks are also $\max \{w_1, \ldots, w_k\}$ and $k + 1$, respectively, the same as the width and depth of the corresponding feedforward network.

## 4 RETHINK THE DEPTH SEPARATION WITH INTRA-LAYER LINKS

Since our focus is the network using ReLU activation and related estimation of the number of pieces, the seminal depth separation theorems closest to us are the following:

**Theorem 1** (Depth separation $k^2$ vs $k$ (Telgarsky, 2015; 2016)). *For a natural number $k \geq 1$, there exists a sawtooth function representable by an $\mathbb{R} \to \mathbb{R}$ $(2k^2 + 1)$-layer feedforward ReLU DNN of width 2 such that if it is also representable by a $(k + 1)$-layer feedforward ReLU DNN, this $(k + 1)$-layer feedforward ReLU DNN should at least have the width of $2^k - 1$.*

**Theorem 2** (Depth separation $k$ vs $k'$ (Arora et al., 2016)). *For every pair of natural numbers $k \geq 1, w \geq 2$, there exists a function representable by an $\mathbb{R} \to \mathbb{R}$ $(k + 1)$-layer feedforward ReLU DNN of width $w$ such that if it is also representable by a $(k' + 1)$-layer feedforward ReLU DNN for any $k' \leq k$, this $(k' + 1)$-layer feedforward ReLU DNN has width at least $\frac{1}{2} w^{\frac{k}{k'}}$.*

Despite being one-dimensional, the above results convincingly reveal that increasing depth can make a ReLU network express a much more complicated function, which is the heart of depth separation. Here, we shed new light on the depth separation problem with intra-layer links. Our primary argument is that if intra-layer links shown in Figure 1(c) are inserted, there exist shallow networks that previously cannot express some hard functions constructed by deep networks now can do the job. Our investigation consists of two parts. First, we substantiate that adding intra-layer links can greatly increase the number of pieces via bound estimation, explicit construction, and functional space analysis. Then, adding intra-layer links can empower the shallow networks to represent complicated functions such as sawtooth functions, without the need of going as wide as before.

### 4.1 INTRA-LAYER LINKS CAN INCREASE THE NUMBER OF PIECES

#### 4.1.1 UPPER BOUND ESTIMATION

**Lemma 3.** *Let $g : \mathbb{R} \to \mathbb{R}$ be a PWL function with $w + 1$ pieces, then the breakpoints of $f := \sigma(g)$ consists of two parts: some old breakpoints of $g$ and at most $w + 1$ newly produced breakpoints. Furthermore, $f$ has $w + 1$ new breakpoints if and only if $g$ has $w + 1$ distinct zero points.*

*Proof.* A direct calculus.  □

**Theorem 4** (Upper bound of feedforward networks). *Let $f : \mathbb{R} \to \mathbb{R}$ be a PWL function represented by an $\mathbb{R} \to \mathbb{R}$ ReLU DNN with depth $k + 1$ and widths $w_1, \ldots, w_k$ of $k$ hidden layers. Then $f$ has at most $\prod_{i=1}^{k} (w_i + 1)$ number of pieces.*

This bound is the univariate case of the bound: $\prod_{i=1}^{k} \sum_{j=0}^{n} \binom{w_i}{j}$, derived in Montúfar (2017) for $n$-dimensional inputs. In Appendix B, we offer constructions to show that this bound is achievable in a depth-bounded but width-unbounded network (depth=3) (Proposition 4) and a width-bounded (width=3) but depth-unbounded network (Proposition 5) in one-dimensional space. Previously many bounds Pascanu et al. (2013); Montufar et al. (2014); Montúfar (2017); Xiong et al. (2020) on linear regions were derived, however, it is unknown that these bounds are vacuous or tight, particularly for networks with more than one hidden layer. What makes Propositions 4 and 5 special is that they for the first time substantiate that Montúfar (2017)'s bound is tight over three-layer and deeper networks, although these results are for the one-dimensional case.

**Remark 1** (Sharpening the bound in (Arora et al., 2016)). Previously, Arora et al. (2016) computed the number of pieces produced by a network of depth $k + 1$ and widths $w_1, \ldots, w_k$ as $2^{k+1} \cdot (w_1 + 1) w_2 \cdots w_k$. The reason why their bound has an exponential term is that when considering how ReLU activation increases the number of pieces, they repetitively computed the old breakpoints generated in the previous layer. Our Lemma 3 implies that the ReLU activation in fact cannot

generate as many as double pieces. Since Arora et al. (2016)'s bound is loose, their depth separation theorem needs to be re-examined.

**Lemma 5.** *Let $g_1, g_2 : \mathbb{R} \to \mathbb{R}$ be two PWL functions with totally $w$ breakpoints. Set $f_1 := \sigma(g_1)$ and $f_2 := \sigma(g_2 - f_1)$. Then the breakpoints of $f_2$ consist of three parts: some breakpoints of $g_2$, some breakpoints of $f_1$, and at most $2w + 2$ newly produced breakpoints. Furthermore, $f_2$ has $2w + 2$ newly produced breakpoints if and only if $g_2 - f_1$ has $2w + 2$ distinct zero points.*

*Proof.* A direct corollary of Lemma 3. $\qquad\square$

Let us illustrate why the intra-linked architecture can produce more pieces. Given two PWL functions $g_1$ and $g_2$ which has totally $w$ breakpoints, in the feedforward architecture, $\sigma(g_1)$ and $\sigma(g_2)$ have totally at most $3w + 2$ breakpoints, which contains at most $w$ old breakpoints of $g_1, g_2$ and at most $2w + 2$ newly produced breakpoints. However, in the intra-linked architecture, $\sigma(g_2 - \sigma(g_1))$ can produce more breakpoints because $\sigma(g_1)$ has two states: activated or deactivated. Then, $\sigma(g_1)$ and $\sigma(g_2 - \sigma(g_1))$ consist of at most $w$ old breakpoints of $g_1, g_2$ and $(w + 1) + (2w + 2) = 3w + 3$ newly produced breakpoints.

**Theorem 6** (Upper bound of intra-linked networks)**.** *Let $f : \mathbb{R} \to \mathbb{R}$ be a PWL function represented by a ReLU DNN with depth $k + 1$, widths $w_1, \ldots, w_k$, and every two neurons linked in each hidden layer as Figure 1(c). Assuming that $w_1, \ldots, w_k$ are even, $f$ has at most $\prod_{i=1}^{k} \left(\frac{3}{2} w_i + 1\right)$ pieces.*

*Proof.* We prove by induction on $k$. For the base case $k = 1$, we assume for every odd $j$, the neurons $\tilde{f}_1^{(j)}$ and $\tilde{f}_2^{(j+1)}$ are linked. The number of breakpoints of $\tilde{f}_1^{(j)}$, $j = 1, \ldots, w_1$, is at most $2 + (-1)^j$. Hence, the first layer yields at most $\frac{3}{2} w_1 + 1$ pieces. For the induction step, we assume that for some $k \geq 1$, any $\mathbb{R} \to \mathbb{R}$ ReLU DNN with every two neurons linked in each hidden layer, depth $k + 1$ and widths $w_1, \ldots, w_k$ of $k$ hidden layers produces at most $\prod_{i=1}^{k} \left(\frac{3}{2} w_i + 1\right)$ pieces. Now we consider any $\mathbb{R} \to \mathbb{R}$ ReLU DNN with every two neurons linked in each hidden layer, depth $k + 2$ and widths $w_1, \ldots, w_{k+1}$ of $k + 1$ hidden layers. By the induction hypothesis, each $\tilde{g}_{k+1}^{(j)}$ has at most $\prod_{i=1}^{k} \left(\frac{3}{2} w_i + 1\right) - 1$ breakpoints. Then the breakpoints of $\sigma(\tilde{g}_{k+1}^{(j)})$ consist of some breakpoints of $\tilde{g}_{k+1}^{(j)}$ and at most $\prod_{i=1}^{k} \left(\frac{3}{2} w_i + 1\right)$ newly generated breakpoints. Then $\tilde{g}_{k+1}^{(j+1)} - \tilde{f}_{k+1}^{(j)}$ has at most $2 \cdot \prod_{i=1}^{k} \left(\frac{3}{2} w_i + 1\right) - 1$ breakpoints, based on Lemma 5. The breakpoints of $\tilde{f}_{k+1}^{(j+1)} = \sigma(\tilde{g}_{k+1}^{(j+1)} - \tilde{f}_{k+1}^{(j)})$ consist of some breakpoints of $\tilde{g}_{k+1}^{(j+1)} - \tilde{f}_{k+1}^{(j)}$ and at most $2 \cdot \prod_{i=1}^{k} \left(\frac{3}{2} w_i + 1\right)$ newly generated breakpoints. Note that $\tilde{g}_{k+1}^{(1)}, \ldots, \tilde{g}_{k+1}^{(w_{k+1})}$ have totally at most $\prod_{i=1}^{k} \left(\frac{3}{2} w_i + 1\right) - 1$ breakpoints. In all, the number of pieces we can therefore get is at most

$$1 + \frac{w_{k+1}}{2} \cdot \left(\prod_{i=1}^{k} \left(\frac{3}{2} w_i + 1\right) + 2 \cdot \prod_{i=1}^{k} \left(\frac{3}{2} w_i + 1\right)\right) + \prod_{i=1}^{k} \left(\frac{3}{2} w_i + 1\right) - 1 = \prod_{i=1}^{k+1} \left(\frac{3}{2} w_i + 1\right).$$

$\qquad\square$

In the following theorems, we offer the bound estimation for high-dimensional cases. The detailed proof for Theorem 8 is put into Appendix A.

**Theorem 7** (Upper Bound of Feedforward Networks (Montúfar, 2017))**.** *Let $f : \mathbb{R}^n \to \mathbb{R}$ be a PWL function represented by an $\mathbb{R}^n \to \mathbb{R}$ ReLU DNN with depth $k + 1$ and widths $w_1, \ldots, w_k$ of $k$ hidden layers. Then $f$ has at most $\prod_{i=1}^{k} \sum_{j=0}^{n} \binom{w_i}{j}$ linear regions.*

**Theorem 8** (Upper Bound of Intra-linked Networks)**.** *Let $f : \mathbb{R}^n \to \mathbb{R}$ be a PWL function represented by an $\mathbb{R}^n \to \mathbb{R}$ ReLU DNN with every two neurons linked in each hidden layer, depth $k + 1$ and widths $w_1, \ldots, w_k$ of $k$ hidden layers. We assume each $w_i$ is even. Then $f$ has at most $\prod_{i=1}^{k} \sum_{j=0}^{n} \binom{\frac{3w_i}{2} + 1}{j}$ linear regions.*

**Remark 2.** Although both adding a new layer (going deep) and adding intra-layer links involve composition, their mechanisms of producing pieces are fundamentally different. While the mechanism of going deep is the repetition effect (multiplication), *i.e.*, the function value of the function being composed is oscillating, and each oscillation can generate corresponding pieces. The mechanism of intra-layer links is the gating effect (addition). The neuron being embedded have two activation states, and each state is leveraged by the neuron being linked to produce a breakpoint. Such a mechanism essentially conforms to the parallelism, which is of width paradigm.

### 4.1.2 EXPLICIT CONSTRUCTION.

Despite that the bound estimation offers some light, to convincingly illustrate that intra-layer links can increase the number of pieces, we need to supply the explicit construction for the intra-linked networks. The number of pieces in the construction should be bigger than either the upper bound of feedforward networks or the maximal number a feedforward network can achieve. Specifically, the constructions for intra-linked networks in Propositions 1 and 2 have a number of pieces larger than the upper bounds of feedforward networks. In Proposition 3, by enumerating all possible cases, we present a construction for an intra-linked network of width 2 and arbitrary depth whose number of pieces is larger than what a feedforward network of width 2 and arbitrary depth possibly achieves.

**Proposition 1** (The bound $\prod_{i=1}^{k} \left( \frac{3w_i}{2} + 1 \right)$ is tight for a two-hidden-layer intra-linked network). *Given an $\mathbb{R} \to \mathbb{R}$ two-hidden-layer ReLU network, with every two neurons linked in each hidden layer, for any even $w_1 \geq 6, w_2 \geq 4$, there exists a PWL function represented by such a network, whose number of pieces is $\left( \frac{3w_1}{2} + 1 \right) \left( \frac{3w_2}{2} + 1 \right)$.*

*Proof.* To guarantee the bound $\prod_{i=1}^{k} \left( \frac{3w_i}{2} + 1 \right)$ is tight, the following two conditions should be satisfied: (i) $\tilde{g}_i^{(j)}$ and $\tilde{g}_i^{(j+1)} - \tilde{f}_i^{(j)}$ have as many zero points as possible so that $\sigma(\tilde{g}_i^{(j)})$ and $\sigma(\tilde{g}_i^{(j+1)} - \tilde{f}_i^{(j)})$ can produce the maximal number of breakpoints; (ii) all old breakpoints of $\left\{ \tilde{g}_i^{(1)}, \ldots, \tilde{g}_i^{(w_i)} \right\}$ are reserved by $\tilde{g}_{i+1}^{(j)}$, an affine transform of $\left\{ \tilde{f}_i^{(1)}, \ldots, \tilde{f}_i^{(w_i)} \right\}$.

We first consider the first hidden layer. Let

$$\tilde{f}_1^{(1)}(x) = \sigma \left( \frac{9}{2}x - 27 \right), \ \tilde{f}_1^{(2)}(x) = \sigma \left( \frac{3}{2}x - \tilde{f}_1^{(1)}(x) \right)$$
$$\tilde{f}_1^{(3)}(x) = \sigma(-2x + 2), \ \tilde{f}_1^{(4)}(x) = \sigma \left( -x + 2 - \tilde{f}_1^{(3)}(x) \right)$$
$$\tilde{f}_1^{(5)}(x) = \sigma \left( -\frac{7}{2}x - \frac{7}{4} \right), \ \tilde{f}_1^{(6)}(x) = \sigma \left( -2x + 8 - \tilde{f}_1^{(5)}(x) \right).$$

When $w_1 = 6$, we set $\tilde{g} = -\frac{2}{9}\tilde{f}_1^{(1)} - \tilde{f}_1^{(2)} + \frac{1}{2}\tilde{f}_1^{(3)} + \tilde{f}_1^{(4)} - \frac{4}{7}\tilde{f}_1^{(5)} - \tilde{f}_1^{(6)}$.

When $w_1 > 6$, for each odd $j > 6$, let $\tilde{f}_1^{(j)} = \sigma \left( -5 \left( x - a_j + 3 \right) \right), \tilde{f}_1^{(j+1)} = \sigma \left( -2 \left( x - a_j \right) - \tilde{f}_1^{(j)} \right)$, where $a_j = -\frac{19}{2} - 9 \left( \frac{j-1}{2} - 3 \right)$, then the output of the first layer is expressed as the following:

$$\tilde{g} = -\frac{2}{9}\tilde{f}_1^{(1)} - \tilde{f}_1^{(2)} + \frac{1}{2}\tilde{f}_1^{(3)} + \tilde{f}_1^{(4)} - \frac{4}{7}\tilde{f}_1^{(5)} - \tilde{f}_1^{(6)} + \sum_{j=7, j \text{ is odd}}^{w_2} (-1)^{\frac{j+1}{2}} \left( \frac{2}{5}f_1^{(j)} + f_1^{(j+1)} \right),$$

which has $\frac{3}{2}w_1 + 1$ pieces and whose adjacent pieces have slopes of opposite signs. Note that any line $y = b$, where $b \in (-13/2, -6)$, can cross all pieces of $\tilde{g} + b$. Thus, $g$ fulfills the conditions of Lemma 5. We divide the breakpoints of $\tilde{g}$ into two parts: $B_{upper} = \{x : x \text{ is a breakpoint of } \tilde{g} \text{ and } \tilde{g}(x) > b\}$ and $B_{lower} = \{x : x \text{ is a breakpoint of } \tilde{g} \text{ and } \tilde{g}(x) \leq b\}$. We refer to their counts as $\#B_{upper}$ and $\#B_{lower}$, respectively.

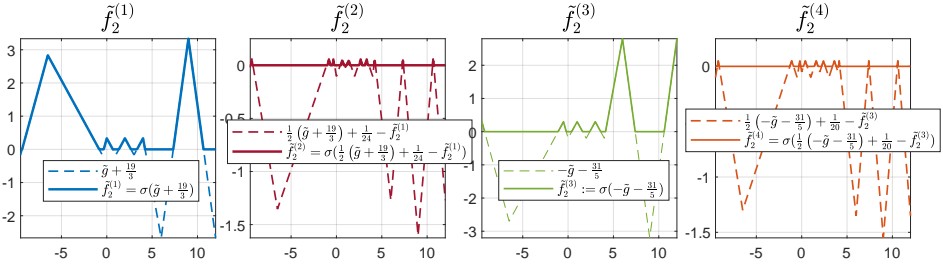

Figure 2: The PWL functions that reach the bound of Proposition 1 when $w_1 = 6, w_2 = 4$.

Next, we construct the second hidden layer. $\tilde{f}_2^{(1)} := \sigma \left( \tilde{g} + b_1 \right)$, where $b_1 \in (-13/2, -6)$, has $\frac{3}{2}w_1 + 1$ new breakpoints. Then by choosing some scaling parameter $a \in (0, 1)$ bias $b_2$ to fulfill Lemma 5, we can also make $a\tilde{g} + b_2 - \tilde{f}_2^{(1)}$ has $3w_1 + 2$ distinct zero-points, which implies $\tilde{f}_2^{(2)} :=$

$\sigma\left(a\tilde{g} + b_2 - \tilde{f}_2^{(1)}\right)$ has $3w_1 + 2$ newly produced breakpoints. Therefore, the affine combination of $\tilde{f}_2^{(1)}$ and $\tilde{f}_2^{(2)}$ contains all breakpoints of $B_{upper}$, and has $\#B_{upper} + \left(\frac{3}{2}w_1 + 1\right) + (3w_1 + 2)$ breakpoints. To reserve all the breakpoints of $\tilde{g}$, we do the similar thing for $-\tilde{g}$ to gain $\tilde{f}_2^{(3)}$ and $\tilde{f}_2^{(4)}$, whose affine combination has $\#B_{lower} + \left(\frac{3}{2}w_1 + 1\right) + (3w_1 + 2)$ breakpoints, which contains all breakpoints in $B_{lower}$, and shares no breakpoints with the affine combination of $\left\{\tilde{f}_2^{(1)}, \tilde{f}_2^{(2)}\right\}$.

Hence, the affine combination of $\left\{\tilde{f}_2^{(1)}, \tilde{f}_2^{(2)}, \tilde{f}_2^{(3)}, \tilde{f}_2^{(4)}\right\}$ has $\#B_{upper} + \# B_{lower} + 2 \cdot \left(\frac{3w_1}{2} + 1\right) + 2 \cdot (3w_1 + 2) = \left(\frac{3w_1}{2}\right) + 6 \cdot \left(\frac{3w_1}{2} + 1\right)$ breaking points, which contains all the breakpoints of $\tilde{g}$. $\left\{\tilde{f}_2^{(1)}, \tilde{f}_2^{(2)}, \tilde{f}_2^{(3)}, \tilde{f}_2^{(4)}\right\}$ are visualized in Figure 2. Repeating this procedure by selecting different $b_1, a, b_2$, we can construct the remaining $\{\tilde{f}_2^{(i)}\}_{i=5}^{w_2}$ such that the affine transformation of $\{\tilde{f}_2^{(i)}\}_{i=1}^{w_2}$ has pieces of

$$\frac{3}{2}w_1 + \frac{3w_2}{2} \cdot \left(\frac{3}{2}w_1 + 1\right) + 1 = \left(\frac{3w_1}{2} + 1\right)\left(\frac{3w_2}{2} + 1\right).$$

$\square$

**Proposition 2** (Use intra-linked networks to achieve a sawtooth function with $\prod_{i=1}^{k}\left(\frac{3w_i}{2}\right)$ pieces). *There exists a $[0,1] \to \mathbb{R}$ function represented by an intra-linked ReLU DNN with depth $k + 1$ and width $w_1, \ldots, w_k$ of $k$ hidden layers, whose number of pieces is at least $\frac{3w_1}{2} \cdot \ldots \cdot \frac{3w_k}{2}$.*

*Proof.* Let $\phi(x) = x$ defined over $[0, \Delta]$. The core of the proof is to use a one-hidden-layer network of $w \geq 2$ neurons to create $\frac{3w}{2}$ pieces from $\phi(x)$.

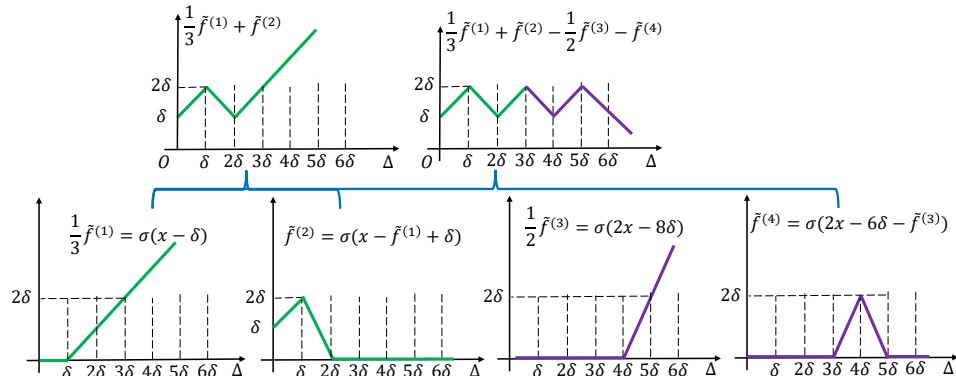

Figure 3: A schematic illustration of how to use an intra-linked network to generate a sawtooth function.

Let $\delta = \frac{2\Delta}{3w}$. Set $\tilde{g}^{(1)} = 3\phi - 3\delta$, $\tilde{f}^{(1)} = \sigma\left(\tilde{g}^{(1)}\right)$, $\tilde{g}^{(2)} = \phi$, $\tilde{f}^{(2)} = \sigma\left(\tilde{g}^{(2)} - \tilde{f}^{(1)} + \delta\right)$, and $\tilde{g}^{(2j+1)} = 4\phi - 4(3j + 1)\delta$, $f^{(2j+1)} = \sigma\left(\tilde{g}^{(2j+1)}\right)$, $\tilde{g}^{(2j+2)} = 2\phi - 6j\delta$, $\tilde{f}^{(2j+2)} = \sigma\left(\tilde{g}^{(2j+2)} - \tilde{f}^{(2j+1)}\right)$, for all $j = 1, \ldots, w/2 - 1$. The output of this one-hidden-layer network is

$$\xi_{\Delta,w}(x) = \frac{1}{3}\tilde{f}^{(1)} + \tilde{f}^{(2)} - \delta + \sum_{j=1}^{\frac{w}{2}-1} (-1)^j \left(\frac{1}{2}\tilde{f}^{(2j+1)} + \tilde{f}^{(2j+2)}\right),$$

which has $\frac{3w}{2}$ pieces on $[0, \Delta]$. $\xi_{\Delta,w}(x)$ is of slope $(-1)^j$ on $[j\delta, (j + 1)\delta]$, $j = 0, \ldots, 3w/2 - 1$, and ranges from 0 to $\delta$ on each piece. Figure 3 shows how the affine transformation of $\{\tilde{f}^{(1)}, \tilde{f}^{(2)}, \tilde{f}^{(3)}, \tilde{f}^{(4)}\}$ constructs a sawtooth function of 6 pieces. Please note that flipping $\phi(x)$ or translating $\phi(x)$ will not prevent $\xi_{\Delta,w}(\phi(x))$ from generating $\frac{3w}{2}$ pieces.

The targeted intra-linked ReLU network with depth $k + 1$ and width $w_1, \ldots, w_k$ of $k$ hidden layers is designed as

$$\xi_{\Delta_k,w_k} \circ \xi_{\Delta_{k-1},w_{k-1}} \circ \cdots \circ \xi_{\Delta_1,w_1}(x), \tag{1}$$

where $\Delta_i = 1/\left( \prod_{j=1}^{i-1} \frac{3w_i}{2} \right)$. $\qquad\square$

**Proposition 3** (Intra-layer links can greatly increase the number of pieces in an $\mathbb{R} \to \mathbb{R}$ ReLU network with width 2 and arbitrary depth)**.** *Let $f : \mathbb{R} \to \mathbb{R}$ be a PWL function represented by an $\mathbb{R} \to \mathbb{R}$ $(k+1)$-layer ReLU DNN with widths $2$ of all $k$ hidden layers. Then number of pieces of $f$ is at most*

$$
\begin{cases}
\sqrt{7}^k, & \text{if $k$ is even,} \\
3 \cdot \sqrt{7}^{k-1}, & \text{if $k$ is odd.}
\end{cases}
$$

*There exists an $\mathbb{R} \to \mathbb{R}$ $(k+1)$-layer $2$-wide ReLU DNN, with neurons linked in each hidden layer, which can produce at least $7 \cdot 3^{k-2} + 2$ pieces.*

*Proof.* The proof is put in Appendix C. $\qquad\square$

### 4.1.3 FUNCTIONAL SPACE ANALYSIS

The above constructive analyses demonstrate that in the maximal sense, intra-layer links can empower a feedforward network to represent a function with more pieces. Now, we move one step forward by showing that intra-layer links can surprisingly expand the functional space of a feedforward network. The reason why this result is surprising is that one tends to think an intra-linked network produces an exclusively different function from a feedforward network. However, here we report that an intra-linked one-hidden-layer network of two neurons can express a feedforward one-hidden-layer network of two neurons (Lemma 9), and the opposite doesn't hold true. Furthermore, given an arbitrary feedforward ReLU network, adding intra-layer links in the first layer can definitely expand its functional space (Theorem 10).

**Lemma 9.** *Let $f^{(1)} = \sigma\left(a^{(1)}x + b^{(1)}\right), f^{(2)} = \sigma\left(a^{(2)}x + b^{(2)}\right), f = c^{(1)}f^{(1)} + c^{(2)}f^{(2)} + d$, where $a^{(1)}a^{(2)} > 0, b^{(1)}, b^{(2)}, c^{(1)}, c^{(2)},$ and $d \in \mathbb{R}$, there exists some $\tilde{a}^{(1)}, \tilde{a}^{(2)}, \tilde{b}^{(1)}, \tilde{b}^{(2)}, \tilde{c}^{(1)}, \tilde{c}^{(2)},$ and $\tilde{d} \in \mathbb{R}$ such that $f = \tilde{c}^{(1)}\tilde{f}^{(1)} + \tilde{c}^{(2)}\tilde{f}^{(2)} + \tilde{d}$, where $\tilde{f}^{(1)} = \sigma\left(\tilde{a}^{(1)}x + \tilde{b}^{(1)}\right), \tilde{f}^{(2)} = \sigma\left(\tilde{a}^{(2)}x + \tilde{b}^{(2)} - \tilde{f}^{(1)}\right).$*

*Proof.* Without loss of generality, we assume $a^{(1)}, a^{(2)} > 0$ and $-\frac{b^{(2)}}{a^{(2)}} < -\frac{b^{(1)}}{a^{(1)}}$. Then $f$ is of slope $0$, $c^{(2)}a^{(2)}$, and $c^{(1)}a^{(1)} + c^{(2)}a^{(2)}$ on $\left(-\infty, -\frac{b^{(2)}}{a^{(2)}}\right], \left[-\frac{b^{(2)}}{a^{(2)}}, -\frac{b^{(1)}}{a^{(1)}}\right]$ and $\left[-\frac{b^{(1)}}{a^{(1)}}, \infty\right)$, respectively. Now we choose $\tilde{a}^{(1)}, \tilde{a}^{(2)}$ satisfying $0 < \tilde{a}^{(1)} < \tilde{a}^{(2)}$, and set $\tilde{b}^{(i)} = \frac{b^{(i)}}{a^{(i)}} \cdot \tilde{a}^{(i)}$, $i = 1, 2$. Then $\tilde{f}^{(1)}$ is of slope $0$ and $\tilde{a}^{(1)}$ on $\left(-\infty, -\frac{b^{(1)}}{a^{(1)}}\right]$ and $\left[-\frac{b^{(1)}}{a^{(1)}}, \infty\right)$, respectively, while $\tilde{f}^{(2)}$ is of slope $0$, $\tilde{a}^{(2)}$, and $\tilde{a}^{(2)} - \tilde{a}^{(1)}$ on $\left(-\infty, -\frac{b^{(2)}}{a^{(2)}}\right), \left[-\frac{b^{(2)}}{a^{(2)}}, -\frac{b^{(1)}}{a^{(1)}}\right]$, and $\left(-\frac{b^{(1)}}{a^{(1)}}, \infty\right)$, respectively. Hence, let $\tilde{c}^{(2)} = c^{(2)}a^{(2)}/\tilde{a}^{(2)}, \tilde{c}^{(1)} = \left(c^{(1)}a^{(1)} + c^{(2)}a^{(2)} - \tilde{c}^{(2)}\left(\tilde{a}^{(2)} - \tilde{a}^{(1)}\right)\right)/\tilde{a}^{(1)}$, and $\tilde{d} = d$, we have $f = \tilde{c}^{(1)}\tilde{f}^{(1)} + \tilde{c}^{(2)}\tilde{f}^{(2)} + \tilde{d}$. $\qquad\square$

**Theorem 10.** *Let $f$ be any $\mathbb{R} \to \mathbb{R}$ PWL representable by a classical $(k+1)$-layer ReLU DNN with widths $w_1 > 2, \ldots, w_k$ of $k$ hidden layers. Then, $f$ can also be represented by a $(k+1)$-layer ReLU DNN with widths $w_1, \ldots, w_k$ of $k$ hidden layers, with neurons in the first layer linked.*

*Proof.* Let the output of the $j$-th neuron of the first layer in the feedforward network be $f_1^{(j)}(x) = \sigma\left(a_1^{(j)}x + b_1^{(j)}\right)$, $j = 1, \ldots, w_1$. Since the feedforward network is invariant to permuting neurons, we can link the arbitrary $j$-th and $j'$-th neuron if $a_1^{(j)}a_1^{(j')} > 0$, which directly concludes the proof according to Lemma 9. $\qquad\square$

### 4.2 MODIFY THE DEPTH SEPARATION THEOREM WITH INTRA-LAYER LINKS

In a broad sense, the depth separation theorem consists of two elements: i) there exists a function representable by a deep network; ii) such a function cannot be represented by a shallow network whose width is lower than a threshold. Since adding intra-layer links can generally improve the capability of a network, if one adds intra-layer links to a shallow network, the function constructed

by a deep network can be represented by a shallow network, even if the width of this shallow network is still lower than the threshold. Theorems 12 and 13 modify the depth separation $k^2$ vs 3 and $k^2$ vs $k$, respectively, by presenting that a shallow network with intra-layer links only needs to go $\frac{2}{3}$ times as wide as before to express the same function.

**Lemma 11** (A network with width=2 can approximate any univariate PWL function (Fan et al., 2018)). *Given a univariate PWL function with $n$ pieces $p(x)$, there exists a $(n+1)$-layer network $\mathbf{D}(x)$ with two neurons in each layer such that $f(x) = \mathbf{D}(x)$.*

**Theorem 12** (Modify the depth separation $k^2$ vs 3). *For every $k \geq 2$, there exists a function $p(x)$ that can be represented by a $(k^2 + 1)$-layer ReLU DNN with 2 nodes in each layer, such that it cannot be represented by a classical 3-layer ReLU DNN $\mathbf{W}_3(x)$ with width less than $k - 1$, but can be represented by a 3-layer, $\frac{2(k-1)}{3}$-wide intra-linked ReLU DNN $\tilde{\mathbf{W}}_3(x)$.*

*Proof.* Combining Theorem 4, Proposition 1, and Lemma 11 straightly concludes the proof. $\square$

**Theorem 13** (Modify the depth separation $k^2$ vs $k$). *For every $k \geq 1$, there is a $[0, 1] \to \mathbb{R}$ PWL function $p(x)$ represented by a feedforward $(2k^2 + 1)$-layer ReLU DNN with at most 6 nodes in each layer, such that it cannot be represented by a classical $(k + 1)$-layer ReLU DNN $W_k(x)$ with width less than $6^k$, but can be represented by a $(k + 1)$-layer intra-linked ReLU DNN $\tilde{W}_k(x)$ with width no more than $4 \cdot 6^{k-1}$.*

*Proof.* Per (Telgarsky, 2016)'s construction, a feedforward $(2k^2 + 1)$-layer ReLU DNN with at most 2 nodes in each layer can produce a sawtooth function of $2^{k^2}$ pieces. Similarly, a feedforward $(2k^2 + 1)$-layer ReLU DNN with at most 6 nodes in each layer can have $6^{k^2}$ pieces. Thus, it follows Theorem 4 that any classical $(k + 1)$-layer ReLU DNN $W_k(x)$ with width less than $6^k - 1$ cannot generate $6^{k^2}$ pieces. However, according to the construction in Proposition 2, let $w_1 = w_2 = \cdots = w_k = 4 \cdot 6^{k-1}$, an intra-linked network can exactly express a sawtooth function with $6^{k^2}$ pieces. $\square$

**Remark 3.** Theorems 12 and 13 implicate that intra-layer links can reduce the bar of the width by 1/3. Although it is not an exponential reduction, our highlight is the existence of such shallow networks that can be transformed by intra-layer links to have the representation power on a par with a deep network. Such shallow networks go against the predictions of depth separation theory. Furthermore, suppose every $n_i$ neurons are intra-linked in the $i$-th layer, the upper bound of the number of pieces by a network of $k$ hidden layers with widths $w_1, \ldots, w_k$ is $\prod_{i=1}^{k} \left( \frac{(n_i+1)w_i}{2} + 1 \right)$. Therefore, by intra-linking more neurons, the bar of the width can be substantially reduced, and more shallow networks will become counterexamples of the depth separation theory.

## 5 DISCUSSION AND CONCLUSION

Please note that intra-layer links are an extension of residual connections. The former is to link the neurons inside a layer, while the latter is to connect neurons across layers. We take the intra-layer links as vertical residual connections, while the shortcuts of ResNet (He et al., 2016) are horizontal residual connections. It is widely recognized that horizontal residual connections can facilitate networks in the dimension of depth, *e.g.*, the residual connection solves the training issues of deep networks and allows the network to go very deep. In contrast, our theoretical results show that vertical residual connections can promote the representation capability of networks in the dimension of width, *e.g.*, the network with intra-layer links does not need to go as wide as before to represent the same function. By leveraging the capabilities of a network in both width and depth domains, we believe that the synergy of horizontal and vertical links in a network will further contribute to more powerful networks. More favorably, both horizontal and vertical links do not incorporate new parameters; therefore, their synergy is likely to enhance model efficiency.

In this draft, via bound estimation, dedicated construction, and functional space analysis, we have shown that a network with intra-layer links is much more expressive than a feedforward one. Then, we have modified the depth separation results to that a shallow network that previously cannot express some functions constructed by deep networks now can do the job with intra-layer links. Our results supplement the existing depth separation theory, and suggest that the potential of wide networks can be released by an appropriate structure. Future endeavors can be put into training wide networks using intra-layer links to achieve comparable performance with deep networks.

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

## A    PROOF OF THEOREM 8

**Lemma 14** (Zaslavsky's Theorem Zaslavsky (1975); Stanley (2004)). *Let $\mathcal{A} = \{H_i \subset V : 1 \leq i \leq m\}$ be an arrangement in $\mathbf{R}^n$. Then, the number of regions for the arrangement $\mathcal{A}$ satisfies*

$$r(\mathcal{A}) \leq \sum_{i=0}^{n} \binom{m}{i}. \tag{2}$$

*Proof.* We prove by induction on $k$. For the base case $k = 1$, $\tilde{f}_1^{(2i-1)} = \sigma\left(\tilde{g}_1^{(2i-1)}\right)$ produces one hyperplane in the input space $\mathbb{R}^n$. Furthermore, $\tilde{f}_1^{(2i)} = \sigma\left(\tilde{g}_1^{(2i)} - \tilde{f}_1^{(2i-1)}\right) = \sigma\left(\tilde{g}_1^{(2i)} - \sigma\left(\tilde{g}_1^{(2i-1)}\right)\right)$ produces at most two hyperplanes in the input space $\mathbb{R}^n$. Therefore, in total, the $w_1$ neurons in the first layer produces $(1+2) \cdot \frac{w_1}{2} = \frac{3w_1}{2}$ hyperplanes in the input space $\mathbb{R}^n$. Then by Zaslavsky's Theorem, it will produce at most $\sum_{j=0}^{n} \binom{\frac{3w_1}{2}+1}{j}$ linear regions in the input space $\mathbb{R}^n$. For the induction step, we assume that for some $k \geq 1$, any $\mathbb{R}^n \to \mathbb{R}$ ReLU DNN with every two neurons linked in each hidden layer, depth $k+1$ and widths $w_1, \ldots, w_k$ of $k$ hidden layers produces at most $\prod_{i=1}^{k} \sum_{j=0}^{n} \binom{\frac{3w_i}{2}+1}{j}$ linear regions. Now we consider any $\mathbb{R}^n \to \mathbb{R}$ ReLU DNN with every two neurons linked in each hidden layer, depth $k+2$ and widths $w_1, \ldots, w_{k+1}$ of $k+1$ hidden layers. Then for each linear region $S$ produced by the first $k+1$ layers, again, $\tilde{f}_{k+1}^{(2i-1)} = \sigma\left(\tilde{g}_{k+1}^{(2i-1)}\right)$ produces one hyperplane in $S$. Furthermore, $\tilde{f}_{k+1}^{(2i)} = \sigma\left(\tilde{g}_{k+1}^{(2i)} - \tilde{f}_{k+1}^{(2i-1)}\right) = \sigma\left(\tilde{g}_{k+1}^{(2i)} - \sigma\left(\tilde{g}_{k+1}^{(2i-1)}\right)\right)$ produces at most two hyperplanes in the $S$. Therefore, in total, the $w_{k+1}$ neurons in the $k+1$ layer produces $(1+2) \cdot \frac{w_{k+1}}{2} = \frac{3w_{k+1}}{2}$ hyperplanes in $S$. Then by Zaslavsky's Theorem, it will produce at most $\sum_{j=0}^{n} \binom{w_{k+1}+1}{j}$ linear regions in $S$. Thus $f$ has at most $\prod_{i=1}^{k} \sum_{j=0}^{n} \binom{\frac{3w_i}{2}+1}{j}$ linear regions.    $\square$

## B    SUPPLEMENTARY RESULTS FOR THE TIGHTNESS OF THEOREM 4

**Proposition 4** (The bound $\prod_{i=1}^{k} (w_i + 1)$ is tight for a depth-bounded but width-unbounded network). *Given an $\mathbb{R} \to \mathbb{R}$ two-hidden-layer ReLU network, for any width $w_1 \geq 3, w_2 \geq 2$ in the first and second hidden layers, there exists a PWL function represented by such a network, whose number of pieces is $(w_1 + 1)(w_2 + 1)$.*

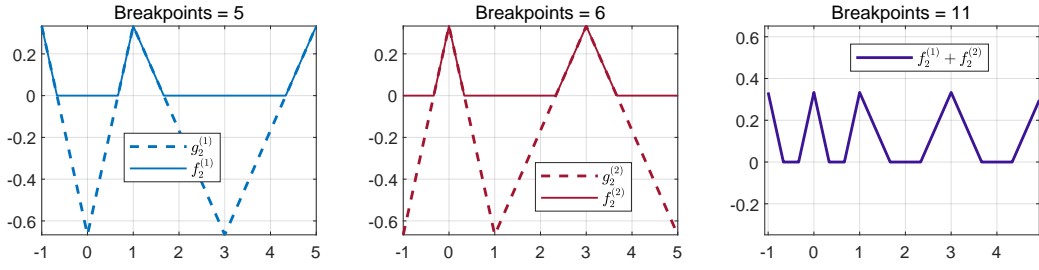

Figure 4: Construction of PWL functions to reach the bound of Proposition 4 when $w_1 = 3, w_2 = 2$.

*Proof.* To guarantee the bound $\prod_{i=1}^{k} (w_i + 1)$ is tight, the following two requirements should be met: (i) each $g_i^{(j)}$, $i = 0, 1, 2, j = 1, \ldots, w_i$, has distinct zero points that are as much as its number of pieces, so that the activation step can produce the most new breakpoints; (ii) the breakpoints of each $g_{(i+1)}^{(j)}$, $i = 0, 1, 2, j = 1, \ldots, w_{i+1}$, as the affine combination of $\left\{f_i^{(1)}, \ldots, f_i^{(w_i)}\right\}$, contains all the breakpoints of $\left\{g_i^{(1)}, \ldots, g_i^{(w_i)}\right\}$, so that all the old breakpoints are reserved.

Now we give the proof in detail. Let $f_1^{(1)}(x) = \sigma(3x), f_1^{(2)}(x) = \sigma(-x + 3), f_1^{(3)}(x) = \sigma\left(\frac{3}{2}x - \frac{3}{2}\right)$. When $w_1 = 3$, we set

$$g_2^{(1)} = -\left(f_1^{(1)} + f_1^{(2)} - f_1^{(3)} - 3 - \frac{1}{w_2+1}\right),$$
$$g_2^{(j)} = f_1^{(1)} + f_1^{(2)} - f_1^{(3)} - 3 - \frac{j}{w_2+1}, j = 2, \ldots, w_2.$$

When $w_1 > 3$, we let $f_1^{(j)} = \sigma(-2x - 2(j-3))$ and

$$g_2^{(1)} = -\left(f_1^{(1)} + f_1^{(2)} - f_1^{(3)} + \sum_{j=4}^{w_1}(-1)^{j-1}f_1^{(j)} - 3 - \frac{1}{w_2+1}\right)$$
$$g_2^{(j)} = f_1^{(1)} + f_1^{(2)} - f_1^{(3)} + \sum_{r=4}^{w_1}(-1)^{r-1}f_1^{(r)} - 3 - \frac{j}{w_2+1}, j = 2, \ldots, w_2.$$

Then $g_2^{(j)}$ has $w_1 + 1$ distinct zero points. Hence for $j = 1, \ldots, w_2$, the breakpoints of $f_2^{(j)} = \sigma\left(g_2^{(j)}\right)$ keeps all breakpoints of $g_2^{(j)}$ and yields $w_1 + 1$ new breakpoints. Note that $f_2^{(j)}$ and $f_2^{(j)}$ do not share new breakpoints, and $f_2^{(1)}$ and $f_2^{(2)}$ covers all the breakpoints of $\left\{g_2^{(j)}\right\}_{j=1}^{w_2}$. Therefore, the total number of pieces via an affine combination of $f_2^{(1)}, \ldots, f_2^{(w_2)}$ is $(w_1 + 1)(w_2 + 1)$ pieces. $\quad\square$

**Proposition 5** (The bound $\prod_{i=1}^{k}(w_i + 1)$ is tight for a width-bounded but depth-unbounded network). *Given an $\mathbb{R} \to \mathbb{R}$ ReLU network with width $w$ for the first layer and 3 for the other layers, for any depth $k \geq 2$, there exists a PWL function represented by such a network, whose number of pieces is $(w + 1) \cdot 4^{k-1}$.*

*Proof.* Let $f_1^{(1)}, \ldots f_1^{(w)}$ be the same as in Proposition 4. Let

$$\tilde{g}_2 = \begin{cases} f_1^{(1)} + f_1^{(2)} - f_1^{(3)} - 3, & \text{if } w = 3, \\ f_1^{(1)} + f_1^{(2)} - f_1^{(3)} + \sum_{j=4}^{w}(-1)^{j-1}f_1^{(j)} - 3, & \text{if } w > 3. \end{cases}$$

We set

$$f_2^{(1)} = \sigma\left(2\tilde{g}_2 - \tfrac{1}{3}\right),$$
$$f_2^{(2)} = \sigma\left(-\tilde{g}_2 + \tfrac{2}{3}\right),$$
$$f_2^{(3)} = \sigma\left(\tfrac{3}{2}\tilde{g}_2 - \tfrac{1}{2}\right).$$

Now we continue our proof by induction. Assume we have constructed $f_i^{(1)}$, $f_i^{(2)}$ and $f_i^{(3)}$, $i \geq 2$. Then we set

$$\tilde{g}_{i+1} = f_i^{(1)} + f_i^{(2)} - f_i^{(3)} - \frac{3}{6^i}$$

and

$$f_{i+1}^{(1)} = \sigma\left(2\tilde{g}_{i+1} - \tfrac{2}{6^i}\right),$$
$$f_{i+1}^{(2)} = \sigma\left(-\tilde{g}_{i+1} - \tfrac{4}{6^i}\right),$$
$$f_{i+1}^{(3)} = \sigma\left(\tilde{g}_{i+1} + \tfrac{3}{6^i}\right).$$

Through a direct calculus, we know $\tilde{g}_{i+1}$ has $(w + 1) \cdot 4^{i-1}$ pieces with opposite slope in every two adjoint pieces and ranges from 0 to $3/6^i$ in each piece except the leftmost and rightmost piece, which implied we can totally obtain $(w + 1) \cdot 4^{k-1}$ pieces. $\quad\square$

## C  PROOF OF PROPOSITION 3

*Proof.* For the first assertion, we claim that each pre-activation $g_i^{(j)}$, $2 \leq i \leq k$, $j = 1, 2$, cannot have its every two adjacent pieces of slope with different sign, which implies the activation cannot produce the most breakpoints as in Lemma 3. In fact, $g_2^{(j)}$, $j = 1, 2$, has at most 3 pieces. If some $g_2^{(j)}$ has 3 pieces, then by exhaustion , we know either it has a 0-slope, or it has two adjacent pieces with slopes of the same sign (see Figure 5). Hence, $f_2^{(j)}$, $j = 1, 2$, has at most 2 newly produced breakpoints. Then the output of the 2-nd layer has at most $2 + 2 \times 2 = 6$ breakpoints, i.e., 7 pieces. Applying the similar method to each piece, we can get our result via a simple induction step.

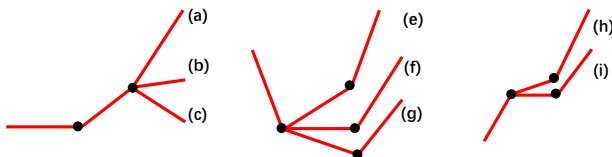

Figure 5: Exhaustion of all possible shapes of $g_2^{(j)}$, $j = 1, 2$ in Proposition 3.

Now we come to the second assertion. For convenience, we say an $\mathbb{R} \to \mathbb{R}$ PWL function $f$ is of "triangle-trapezoid-triangle" shape on $[a, b] \subset \mathbb{R}$, if there exists a partition of $[a, b] : a < x_1 < x_2 < \cdots < x_6 < b$ and a positive constant $c$, such that

$$f|_{[a,b]}(x) = \begin{cases} c & \text{, if } x = a, x_2, x_5, b \\ -c & \text{, if } x = x_1, x_6 \\ -3c & \text{, if } x \in [x_3, x_4] \\ \text{linear connection} & \text{, otherwise.} \end{cases}$$

Given a PWL function $f : \mathbb{R} \to \mathbb{R}$ of "triangle-trapezoid-triangle" shape on $[a, b]$, with corresponding partition $a < x_1 < x_2 < \cdots < x_6 < b$ and $f(a) = c > 0$, if we set

$$\begin{aligned} g^{(1)} &= 4f \\ g^{(2)} &= 2f - \frac{3c}{2} \\ f^{(1)} &= \sigma\left(g^{(1)}\right) \\ f^{(2)} &= \sigma\left(g^{(2)} - f^{(1)}\right), \end{aligned}$$

then $g = -\frac{1}{4}f^{(1)} + f^{(2)} + \frac{c}{8}$ is of "triangle-trapezoid-triangle" shape on $[a, x_2]$, $[x_2, x_5]$ and $[x_5, b]$ respectively.

Using this fact, we can construct a PWL function represented by a $(k+1)$-layer 2-wide intra-linked ReLU DNN, which has $7 \cdot 3^{k-2} + 2$ pieces. Actually, set

$$\begin{aligned} \tilde{f}_1^{(1)} &= \sigma(2x), \\ \tilde{f}_1^{(2)} &= \sigma\left(x - \tilde{f}_1^{(1)} + 1\right), \\ \tilde{g}_2^{(1)} &= -4\tilde{f}_1^{(2)} + 2, \\ \tilde{g}_2^{(2)} &= -2\tilde{f}_1^{(2)} + \frac{3}{2}, \end{aligned}$$

then through a direct calculus, $\frac{1}{4}\tilde{f}_2^{(1)} + \tilde{f}_2^{(2)} - \frac{3}{8}$ is of "triangle-trapezoid-triangle" shape on $[-1, 1]$. Using the fact above repeatedly, we can construct a PWL function represented by a $\mathbb{R} \to \mathbb{R}$ $(k+1)$-layer, 2-wide, intra-linked ReLU DNN, which is constant on $(-\infty, -1] \cup [1, \infty)$ and of "triangle-trapezoid-triangle" shape on $\left[-1 + \frac{2n}{3^{k-2}}, -1 + \frac{2(n+1)}{3^{k-2}}\right]$, $n = 0, \ldots, 3^{k-2} - 1$. $\qquad \square$

## D  VALIDATING THE REPRESENTATION POWER OF INTRA-LINKED LINKS

Inspired by the encouraging theoretical analyses, we validate whether or not the intra-layer links can assist a network to deliver superior performance in real-world tasks. The task is to predict if a credit card holder will get churned so that the bank can provide better service to turn holders' decisions. This prediction task has 10,000 raw samples, and each has 18 customers' portfolio features including age, salary, marital status, credit card limit, credit card category, etc. The labels are 'get churned' or 'stay'. The detailed description of data and this task can be referred to in Kaggle[1].

The data are preprocessed as follows: The discrete value is assigned to different education levels based on the mapping { 'Uneducated': 0, 'High School': 1, 'College': 2, 'Graduate': 3, 'Post-Graduate': 4, 'Doctorate': 5 }. The income situation is assigned with values based on the mapping: { 'Less than \$40K': 0, '\$40K - \$60K': 1, '\$60K - \$80K': 2, '\$80K - \$120K': 3, '\$120K +': 4 }.

---

[1] https://www.kaggle.com/datasets/whenamancodes/credit-card-customers-prediction

The female and male are mapped to 0 and 1, respectively. All samples with missing attributions are deleted. Finally, the processed data have 7,081 data points. Then, the data are randomly split into training and testing sets with a ratio of 0.8:0.2.

We build networks with intra-layer links and compare them with the corresponding feedforward networks without intra-layer links. The optimizer is Adam Kingma & Ba (2014) with a learning rate of 0.1. The loss function is the binary cross-entropy function. The evaluation metric is the widely used micro-F1 score. The prediction results by feedforward and intra-linked networks are summarized in Table 1, from which we draw two highlights. First, when the same structure is used, the intra-linked network consistently outperforms the feedforward network. More favorably, the intra-linked network takes the lead by a large margin: the minimum gain is 1.68% for the network structure 2-32-2, while the maximum gain is 2.66% for the network structure 2-16-16-2. Such a superiority corroborates our theoretical analysis that adding intra-layer links can boost the network's representation power. Second, the one-hidden-layer intra-linked network can sometimes perform superbly over the two-hidden-layer feedforward network (22-16-2 vs 22-8-8-2, 22-32-2 vs 22-16-16-2, 22-64-2 vs 22-32-32-2), when their parameters are comparable. This phenomenon suggests that the representation power of an intra-linked network is different from that of a feedforward network with more layers.

Table 1: Credit card customers prediction by feedforward and intra-linked networks.

| Feedforward | #Parameters | Micro-F1(%) | Intra-linked | #Parameters | Micro-F1(%) |
|---|---|---|---|---|---|
| 22-8-2 | 202 | 78.24±0.23% | 22-8-2 | 202 | **80.13±0.37%** |
| 22-16-2 | 402 | 78.93±0.19% | 22-16-2 | 402 | **80.83±0.41%** |
| 22-32-2 | 802 | 80.53±0.18% | 22-32-2 | 802 | **82.21±0.27%** |
| 22-64-2 | 1602 | 82.03±0.23% | 22-64-2 | 1602 | **83.52±0.34%** |
| 22-8-8-2 | 274 | 80.21±0.32% | 22-8-8-2 | 274 | **82.11±0.42%** |
| 22-16-16-2 | 674 | 81.14±0.34% | 22-16-16-2 | 674 | **83.80±0.37%** |
| 22-32-32-2 | 1858 | 83.14±0.28% | 22-32-32-2 | 1858 | **84.79±0.45%** |
| 22-64-64-2 | 5762 | 84.35±0.24% | 22-64-64-2 | 5762 | **85.44±0.25%** |

# E    INSERTING INTRA-LAYER LINKS VS STACKING LAYERS

Here, we argue that adding intra-layer links is not equivalent to increasing a new layer in the following three aspects:

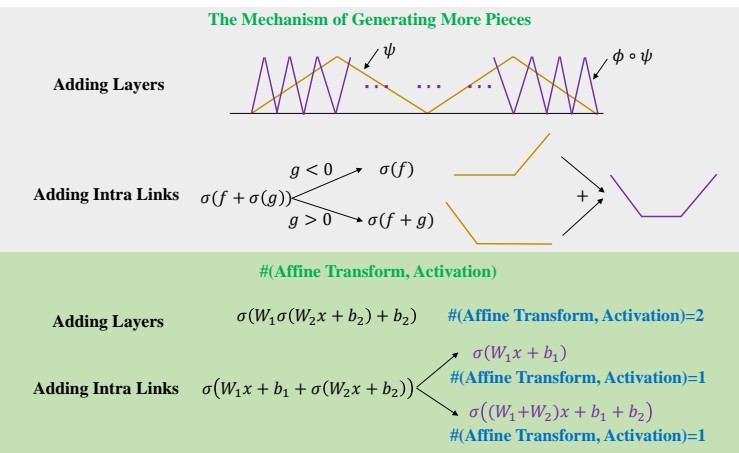

Figure 6: Adding intra-layer links is not equivalent to increasing depth in terms of the mechanism of generating more pieces, the number of (affine transform, activation), and function classes.

- As Figure 6 shows, their mechanisms of producing pieces are fundamentally different. While the mechanism of adding a new layer is the repetition effect (multiplication), *i.e.*, the function value of the function being composed is oscillating, and each oscillation can generate more pieces. The mechanism of intra-layer links is the gating effect (addition). The neuron being embedded have two activation states, and each state is leveraged by the

neuron being linked to produce a breakpoint. Two states are integrated to generate more pieces.

- Although both stacking a new layer and adding intra-layer links involve composition, they involve different numbers of (affine transform, activation). In a feedforward network, adding a new layer means the depth increases by 1, and the characteristic of stacking a new layer is doing the affine transformation followed by activation. As Figure 6 illustrates, a feedforward network with two layers involves two times of (affine transformation, activation). In contrast, adding intra-layer links in a fully-connected layer actually exerts a gating effect. When $\sigma(W_2 x + b_2) > 0$, the output is $\sigma((W_1 + W_2)x + b_1 + b_2)$; when $\sigma(W_2 x + b_2) = 0$, the output is $\sigma(W_1 x + b_1)$. The number of (affine transformation, activation) is still one for both cases.

- The function classes (the set of functions represented by some given neural network architecture) of our intra-linked network and the deeper feedforward network are not the same, and this will make a big difference. In some sense, the deeper feedforward network has a larger function class, and the function class of our intra-linked network is just a subset of it. However, our intra-linked network has more expressive power (i.e., number of pieces, VC dimension) per parameter. Also, the experiments showed that intra-linked networks can achieve better accuracy in solving real-world problems. This phenomenon can be seen as an analog to the comparison between CNNs and fully-connected NNs. The function classes of CNNs are just subsets of the function classes of fully-connected NNs with some further restrictions on the weights. However, CNNs usually have more expressive power per parameter and achieve better results in practice.

Topologically, one can define the depth of a network as the length of the longest path from the input to the output, by regarding a neural network as a directed acyclic graph. With this definition, adding the intra-layer links certainly increases the depth by roughly a factor of two. However, if we examine the operations along the longest path, based on the above second observation, the number of (affine transformation, activation) remains to be the same with the number of layers. Thus, if the depth is defined as the number of (affine transformation, activation) that are actually executed, the depth of the intra-linked network is the same as that of the feedforward network. Since topologically one can make any shallow network deep by making a series of reducible identity layers, we argue that the intrinsic computational operation is more important than the extrinsic topology for the depth separation theory.

## F    EXTENSION TO MORE INTRA-LAYER LINKS

For an $\mathbb{R}^{w_0} \to \mathbb{R}^{w_{k+1}}$ ReLU DNN with depth $k + 1$, widths $w_1, \ldots, w_k$ of $k$ hidden layers, We now assume that every $n_i$ neurons are intra-layer linked, where $n_i$ can divide $w_i$ without remainder. We use $\tilde{\mathbf{f}}_0 = \left[ \tilde{f}_0^{(1)}, \ldots, \tilde{f}_0^{(w_0)} \right] = \mathbf{x} \in \mathbb{R}^{w_0}$ to denote the input of the network. Let $\tilde{\mathbf{f}}_i = \left[ \tilde{\mathbf{f}}_i^{(1)}, \ldots, \tilde{\mathbf{f}}_i^{(w_i)} \right] \in \mathbb{R}^{w_i}$, then for $i = 1, \ldots, k, j = 1, \ldots, w_i$, we use $\tilde{g}_i^{(j)} = \left\langle \tilde{\mathbf{a}}_i^{(j)}, \tilde{\mathbf{f}}_{i-1} \right\rangle + \tilde{b}_i^{(j)}$ to denote the $j$-th pre-activation in the $i$-th layer respectively, where $\mathbf{a}_i^{(j)} \in \mathbb{R}^{w_{i-1}}, b_i^{(j)} \in \mathbb{R}$ are some parameters. In an intra-linked network, the $j$ -th, ..., $(j + n_i - 1)$-th neurons in the $i$-th layer are linked, and the $(j + n_i)$-th, $\cdots$, $(j + 2n_i - 1)$-th neurons in the $i$-th layer are linked. We prescribe $\tilde{f}_i^{(j)} = \sigma\left( g_i^{(j)} \right)$ and $\tilde{f}_i^{(j+l)} = \sigma\left( g_i^{(j+l)} - \tilde{f}_i^{(j+l-1)} \right)$, for $l = 1, \ldots, n - 1$. The output of the network is $\tilde{g}_{k+1}^{(j)} = \left\langle \tilde{\mathbf{a}}_k^{(j)}, \tilde{\mathbf{f}}_k \right\rangle + \tilde{b}_k^{(j)}, j = 1, \ldots, w_{k+1}$, where $\tilde{\mathbf{a}}_k^{(j)} \in \mathbb{R}^{w_k}, \tilde{b}_k^{(j)} \in \mathbb{R}$ are parameters.

**Theorem 15.** *Let $f : \mathbb{R} \to \mathbb{R}$ be a PWL function represented by a $\mathbb{R} \to \mathbb{R}$ ReLU DNN with depth $k + 1$, widths $w_1, \ldots, w_k$ of $k$ hidden layers and every $n_i$ neurons linked in the $i$-th hidden layer for some positive integer $n_i$ that divides $w_i$ without remainder. Then $f$ has at most $\prod_{i=1}^{k} \left( \frac{n_i + 1}{2} w_i + 1 \right)$ pieces.*

*Proof.* For convenience, we assume in the $i$-th layer, the $j$-th, $\cdots$, $(j + n_i - 1)$-th neurons are linked, for $i = 1, \ldots, k, j = 1, \ldots, \bar{w}_i - 1$. For the first layer, $\tilde{f}_1^{(1)}$ has one breakpoint and each $\tilde{f}_1^{(j)}$ has at most $j$ newly produced breakpoints and some old breakpoints of $\tilde{g}_1^{(j)}$ and $\tilde{f}_1^{(j-1)}$, for

$j = 2, \ldots, n_1$. Hence, the first layer gives at most $\frac{n_i+1}{2}w_i + 1$ pieces. Then the rest of the proof is similar to Theorem 6. □

**Corollary 15.1.** *Let $f : \mathbb{R} \to \mathbb{R}$ be a PWL function represented by a $\mathbb{R} \to \mathbb{R}$ ReLU DNN with all the neurons linked in each hidden layer, depth $k+1$, and widths $w_1, \ldots, w_k$ of $k$ hidden layers. Then $f$ has at most $\prod_{i=1}^{k}\left(\frac{w_i+1}{2}w_i + 1\right)$ pieces.*

**Theorem 16** (The bound $\prod_{i=1}^{k}\left(\frac{(w_i+1)w_i}{2} + 1\right)$ is tight for a one-hidden-layer intra-linked network). *Given an $\mathbb{R} \to \mathbb{R}$ one-hidden-layer ReLU network with all neurons linked in the hidden layer, there exists a PWL function represented by such a network, whose number of pieces is $\frac{(w_1+1)w_1}{2} + 1$.*

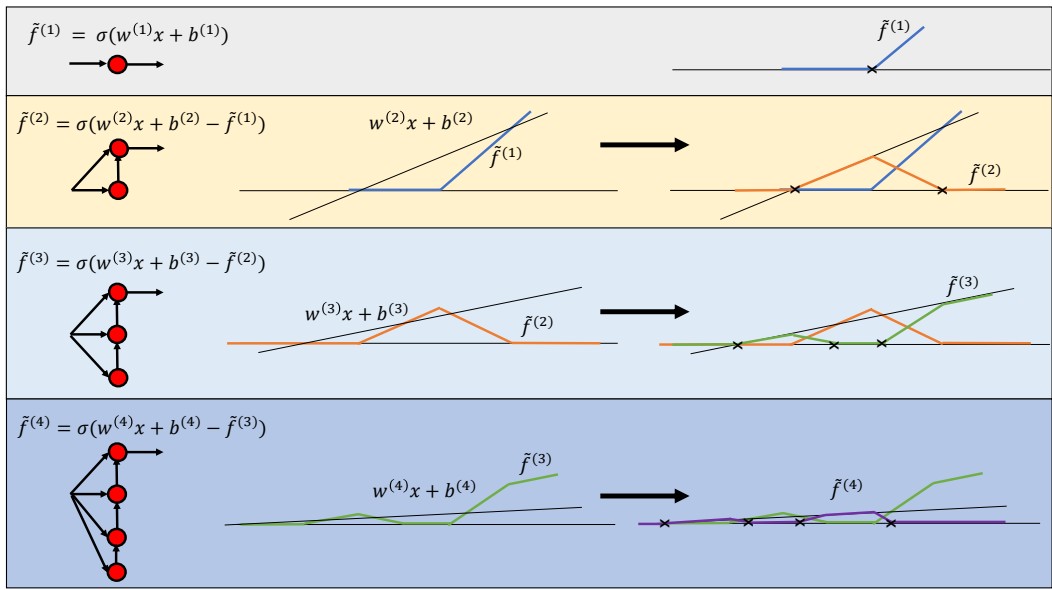

Figure 7: The construction demonstrating that the bound $\prod_{i=1}^{k}\left(\frac{(w_i+1)w_i}{2} + 1\right)$ is tight for a one-hidden-layer intra-linked network.

*Proof.* Without loss of generality, a one-hidden-layer network with all neurons intra-linked is mathematically formulated as the following:

$$\begin{cases} \tilde{f}^{(1)} = \sigma(w^{(1)}x + b^{(1)}) \\ \tilde{f}^{(j+1)} = \sigma(w^{(j)}x + b^{(j)} - \tilde{f}^{(j)}) \end{cases}. \tag{3}$$

To prove that the bound $\prod_{i=1}^{k}\left(\frac{(w_i+1)w_i}{2} + 1\right)$ is tight for a one-hidden-layer network, the key is to make each $\tilde{f}^{(j)}$ produce $j$ new breakpoints and have $j$ non-zero pieces that share a point with $y = 0$. We use mathematical induction to derive our construction. Figure 7 schematically illustrates our construction idea.

First, let $\tilde{f}^{(1)} = \sigma(x+1)$ and $\tilde{f}^{(2)} = \sigma(0.5 \times (x+2) - \tilde{f}^{(1)})$. Note that $\tilde{f}^{(1)}$ has 1 non-zero piece that shares a point with $y = 0$, and $\tilde{f}^{(2)}$ has 2 non-zero pieces that share a common point with $y = 0$.

Then, given $\tilde{f}^{(j)}, j \geq 2$, we suppose $\tilde{f}^{(j)}$ has $j$ non-zero pieces that share a point with $y = 0$. Since $\tilde{f}^{(j)}$ is continuous, we select its peaks $\{(x_{p_i}, \tilde{f}^{(j)}(x_{p_i}))\}$ by the following conditions: i) $\tilde{f}^{(j)}$ is not differentiable at $x_{p_i}$; ii) $\tilde{f}^{(j)}(x_{p_i}) \neq 0$. Next, let $(x^*, \tilde{f}^{(j)}(x^*))$ be the lowest peak of $\tilde{f}^{(j)}$. As long as the slope $w^{(j+1)}$ and the bias $b^{(j+1)}$ satisfy

$$\begin{cases} w^{(j+1)} < \frac{\tilde{f}_1^{(j)}(x^*)}{x^*+j+1} \\ b^{(j+1)} = w_{j+1} \times (j+1) \end{cases}, \tag{4}$$

$w^{(j+1)}x + b^{(j+1)}$ crosses and only crosses $j$ pieces of $\tilde{f}^{(j)}$. These pieces are exactly non-zero pieces that share a point with $y = 0$. Thus, plus the breakpoint $-\frac{b^{(j+1)}}{w^{(j+1)}}$, $\tilde{f}^{(j+1)}$ generates a total of $j + 1$ new breakpoints. At the same time, $\tilde{f}^{(j+1)}$ has $j + 1$ non-zero pieces that share a point with $y = 0$. Figure 7 illustrates the process of induction.

Finally, the total number of breakpoints is $\sum_{j=1}^{w_1} j = \frac{(w_1+1)w_1}{2}$, which concludes our proof.

$\square$

**Theorem 17** (An arbitrarily deep network of width=4 and with all neurons in each layer intra-linked can achieve at least $9^k$ pieces). *There exists an $\mathbb{R} \to \mathbb{R}$ function represented by an intra-linked ReLU DNN with depth $k$, width $4$ in each layer, and all neurons in each layer intra-linked, whose number of pieces is at least $9^k$.*

*Proof.* The core of the proof is to use a one-hidden-layer all-neuron-intra-linked network of width 4 to create a quasi-sawtooth function with as many pieces as possible. We construct four neurons as follows:

$$\begin{cases} \tilde{f}^{(1)} = \sigma(2x) \\ \tilde{f}^{(2)} = \sigma(x + 1 - \sigma(\tilde{f}^{(1)})) \\ \tilde{f}^{(3)} = \sigma(\frac{1}{3}(x+2) - \tilde{f}^{(2)}) \\ \tilde{f}^{(4)} = \sigma(\frac{1}{9}(x+3) - \tilde{f}^{(3)}) \end{cases}. \tag{5}$$

The profiles of $\tilde{f}^{(1)}, \tilde{f}^{(2)}, \tilde{f}^{(3)}, \tilde{f}^{(4)}$ are shown in Figure 8(a).

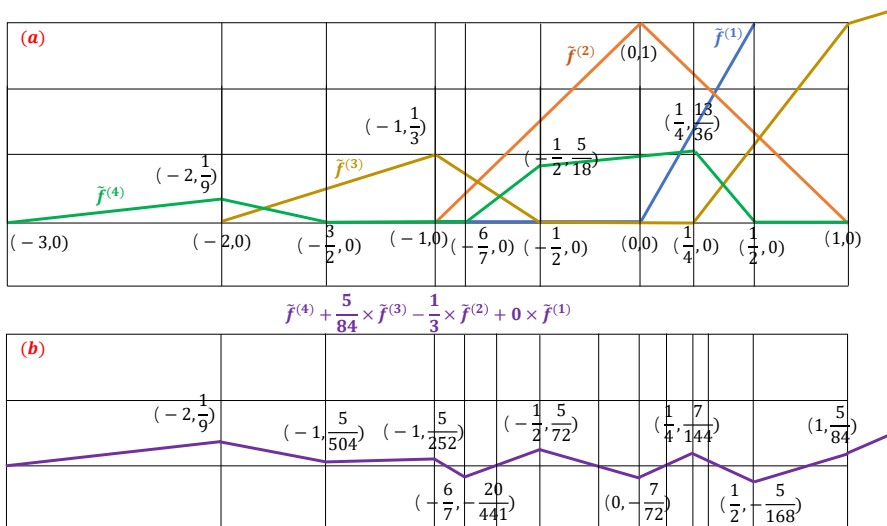

Figure 8: A schematic illustration of how to use an intra-linked network to generate a sawtooth function.

By combining $\tilde{f}^{(1)}, \tilde{f}^{(2)}, \tilde{f}^{(3)}, \tilde{f}^{(4)}$ with carefully calibrated coefficients, we have the following quasi-sawtooth function that has 9 pieces are

$$\eta(x) = \tilde{f}^{(4)} + \frac{5}{84} \times \tilde{f}^{(3)} - \frac{1}{3} \times \tilde{f}^{(2)} + 0 \times \tilde{f}^{(1)}. \tag{6}$$

As shown in Figure 8(b), we have marked all breakpoints of $\eta(x)$ to validate its correctness.

Next, we just need to let each layer of the intra-linked network represent a stretched and down-pulled variant of $\eta(x)$, *e.g.*, the $k$-th layer $\eta_k(x) = M_k \cdot \eta(x) - B_k$, where $M_k$ is a sufficiently large number and $B_k > \frac{5}{504}M_k + 3$ to ensure that $[-3, 0]$ is within the function range of $\eta_k(x)$.

Finally, the constructed network is

$$\eta_k \circ \eta_{k-1} \circ \cdots \circ \eta_1(x). \tag{7}$$

$\square$

**Theorem 18** (An arbitrarily deep network of width=3 and with all neurons in each layer intra-linked can achieve at least $5^k$ pieces). *There exists an $\mathbb{R} \to \mathbb{R}$ function represented by an intra-linked ReLU DNN with depth $k$, width 3 in each layer, and all neurons intra-linked in each layer, whose number of pieces is at least $5^k$.*

*Proof.* Following the same spirit in proof of Theorem 17, we construct three neurons as follows:

$$\begin{cases} \tilde{f}^{(1)} = \sigma(2x) \\ \tilde{f}^{(2)} = \sigma(x + 1 - \sigma(\tilde{f}^{(1)})) \\ \tilde{f}^{(3)} = \sigma(\frac{1}{3}(x+2) - \tilde{f}^{(2)}) \end{cases}. \tag{8}$$

The target function that returns us 5 pieces is

$$\xi(x) = \frac{1}{100} \times \tilde{f}^{(3)} - \frac{1}{3} \times \tilde{f}^{(2)} + 0 \times \tilde{f}^{(1)}. \tag{9}$$

Next, we just need to let each layer of the intra-linked network represent a stretched and down-pulled variant of $\xi(x)$, *e.g.*, the $k$-th layer $\xi_k(x) = T_k \cdot \eta_k(x) - C_k$, where $T_k$ is a sufficiently large number and $C_k > \frac{1}{200}T_k + 2$ to ensure that $[-2, 0]$ is within the function range of $\xi_k(x)$.

Finally, the constructed network is

$$\xi_k \circ \xi_{k-1} \circ \cdots \circ \xi_1(x). \tag{10}$$

$\square$

**Remark 4.** Suppose each layer has $w$ neurons: $w_1 = w_2 = \cdots = w_k = w$, and $n = w$, the upper bound of the intra-layer linked network is $(\frac{(w+1)w}{2} + 1)^k$, which approximately equals to that of a feedforward network with width $w_1 = w_2 = \cdots = w_k = \frac{(w+1)w}{2}$. At this time, the improvement of representation power by intra-links is $\mathcal{O}(w)$ instead of a constant-level improvement. Thus, the separation is still valid if one allows increasing the width of feedforward networks by a constant factor.

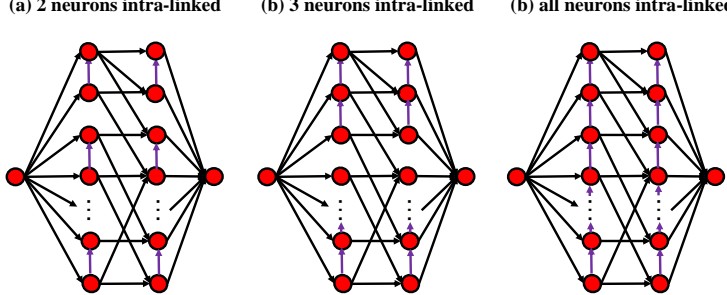

**(a) 2 neurons intra-linked**    **(b) 3 neurons intra-linked**    **(b) all neurons intra-linked**

Figure 9: The improvement of representation power by intra-links is $\mathcal{O}(w)$ when all neurons in a layer are intra-linked.

**Proposition 6** (Modify the depth separation $k^2$ vs 2). *For every $k \geq 2$, there exists a function $p(x)$ that can be represented by a $(k^2 + 1)$-layer ReLU DNN with 2 nodes in each layer, such that it cannot be represented by a classical 2-layer ReLU DNN $\mathbf{W}_2(x)$ with width less than $k^2 - 1$, but can be represented by a 2-layer, $(2k)$-wide intra-linked ReLU DNN $\tilde{\mathbf{W}}_2(x)$.*

*Proof.* Combining Theorem 4, Theorem 16, and Lemma 11 straightly concludes the proof. $\square$

## G  ANALYSIS EXTENDED TO ONE-NEURON-WIDE RESNET

Our analysis can be extended to ResNet to show the power of residual connections. Let us use a one-neuron-wide ResNet to demonstrate this point. It is straightforward to see that a one-neuron-wide ReLU DNN can represent PWL functions with at most three pieces, no matter how deep the network is. However, if we add residual connections to the network, which gives a ResNet, it can represent PWL functions with much more pieces.

**Theorem 19.** *Let $f : \mathbb{R} \to \mathbb{R}$ be a PWL function represented by a one-neuron-wide ResNet. Mathematically, $f = c_{k+1}f_k + g_k$, where $g_1(x) = x$, $f_i = \sigma(a_ig_i + b_i)$, $g_{i+1} = c_if_i + g_i$, $c_{k+1}, a_i, b_i, c_i$ are parameters, for $i = 1, \ldots, k$. Then $f$ has at most $2k + 2$ pieces.*

*Proof.* The first claim follows from Lemma 3 and a simple induction step. Following the idea of the construction in Propositions 1 and 2, we set $c_i = -2$ and $a_i = 1$ for all $i$ and set $b_1 = 0$, $b_i = 2 - 2^{-i+2}$ for $i = 2, \ldots, k$ . □

Theorem 19 confirms that adding simple links can greatly improve the representation ability of a network. Actually, both ResNet and intra-layer linked networks do not increase the number of parameters, but they can represent more complicated functions than the feedforward of the same neurons used in each layer. Hence, the linked structure can improve the efficiency of parameters. Besides, we can see from the proof of Theorem 19 that the idea and construction in analyzing intra-linked networks can indeed be utilized to analyze other architecture of networks.

