# OpenReview forum: "Rethink Depth Separation with Intra-layer Links"
_ICLR.cc/2023/Conference — Submitted to ICLR 2023_

### Official Review · Reviewer_p5Vt · 2022-10-24

**Confidence:** 4
**Correctness:** 2
**Technical Novelty And Significance:** 3
**Empirical Novelty And Significance:** Not applicable
**Recommendation:** 6

**Clarity, Quality, Novelty And Reproducibility:**

Clarity:
- The paper is overall well-written and easy to follow.
- Minor question: in the end of remark 1, I was wondering if further elaboration could be made to clarify what exactly “needs to be re-examined”.

Quality:
- The theoretical results are clearly stated with full proof provided in the appendix.


Novelty:
- Study the representation power of deep networks with different architectures is an interesting direction. The intra-linked networks studied in this paper seems to be new in the literature.


Reproducibility:
- This is a theoretical work so there are no experiments to reproduce. Full proofs are provided in the appendix, but I didn’t check them.


**Strength And Weaknesses:**

Strength:
- Studying the depth-separation theory with different architecture is an interesting direction, which may help us to have better understanding of different architectures.
- The paper is well-motivated and overall is easy to follow.
- The setup of the theoretical framework is clearly stated, some of the proof ideas are discussed and the proofs are provided.


Weaknesses:
- The studied intra-linked networks are different from commonly used network architectures like feedforward networks and ResNet-type networks with shortcuts. It is not clear to me whether such analysis could be used on more commonly used networks in practice.
- The improvement using intra-linked networks over standard feedforward networks seems to be not very strong in my opinion. For example,
  - (1) the results in Section 4.4.1 are all upper bounds on the number of linear pieces for ReLU networks with or without intra-links. Since they are only upper bounds, I don’t think these could give a valid separation. Moreover, even these upper bounds are tight, the separation seems to be not strong, since we are comparing $\Pi_i (3w_i/2 +1)$ with $\Pi _i(w_i +1)$ (the number of linear pieces), which means it is possible that a slightly wider feedforward network has the same representation power as network with intra-links.
  - (2) In Section 4.1.2, the separation only applies to feedforward ReLU networks and ReLU networks with intra-links under the *exact same* depth and width. It is not clear to me whether the separation is still valid if one allows to increase the width of feedforward networks by a constant factor (say 2). I currently tend to believe that the separation does not hold in such cases, which makes me to view the separation not strong enough. Please correct me if I am wrong.
  - (3) In Section 4.1.3, it is only shown that every standard feedforward ReLU networks can also be represented with a network with intra-links and same depth and width. I would not view this as a valid separation result.
  - (4) In Section 4.2, Theorem 12 and 13 show that the improvement on width from using feedforward networks to intra-linked networks are from $k-1$ to $2(k-1)/3$ and from $6^k$ to $4*6^{(k-1)}$. These improvement does not significantly change the representation power of networks in the sense of the order remains to be $O(k)$ or $e^{O(k)}$.


**Summary Of The Paper:**

In this paper, authors revisited the depth separation theory of deep neural networks. While most of the previous paper focused on feedforward networks, this paper studied feedforward networks with intra-layer links. The authors showed that such ReLU networks with intra-layer links can increase its representation power compared with standard ReLU feedforward networks, in the sense that the required width to represent certain function could be reduced.

**Summary Of The Review:**

In summary, this paper studied the representation power of the intra-linked deep networks and showed certain separation results with standard ReLU networks, which I believe is an interesting result. However, as mentioned in the weakness part, the separation results do not seem to be very strong, which is my main concern. Therefore, I’m currently leaning towards rejection.

Update after authors' response: I'm willing to increase my score after authors' response.

---

> ### Author Response · Authors · 2022-11-08
> **Response to Reviewer p5Vt**
>
> Dear Reviewer p5Vt:
>
> We would like to thank you for your recognition of the novelty and significance of our work and the wonderful comments that motivate us to further enhance our draft. We have extended our analysis to ResNet-type networks to show the power of residual connections (please refer to [Q1]). We also addressed your concerns on the constant-level improvement by the intra-layer links (please refer to [Q2]) in the rebuttal. Please feel free to let us know if you have additional comments, and we are always happy to discuss them here.
>
> **[Q1] The studied intra-linked networks are different from commonly used network architectures like feedforward networks and ResNet-type networks with shortcuts. It is not clear to me whether such analysis could be used on more commonly used networks in practice.**
>
> Our analysis indeed can be extended to ResNet to show the power of residual connections. Let us use a one-neuron-wide ResNet to demonstrate this point. It is straightforward to see that a one-neuron-wide ReLU DNN can represent PWL functions with at most three pieces, no matter how deep the network is. However, if we add residual connections to the network, which gives a ResNet, it can represent PWL functions with much more pieces. We have derived the following theorem, and the proof can be referred to in Section G of the Appendix.
>
> (Theorem) *Let $f: \mathbb{R} \rightarrow \mathbb{R}$ be a PWL function represented by a one-neuron-wide ReLU ResNet. Mathematically, $f=c_{k+1} f_{k}+g_{k}$, where $g_{1}(x)=x, f_{i}=\sigma\left(a_{i} g_{i}+b_{i}\right), g_{i+1}=c_{i} f_{i}+g_{i}, c_{k+1}, a_{i}, b_{i}, c_{i}$  are parameters, for  $i=1, \ldots, k$. Then $f$ has at most $2k+2$  pieces.*
>
> **[Q2] The improvement using intra-linked networks over standard feedforward networks seems to be not very strong in my opinion.**
>
> - Adding more intra-layer links can boost the gain from the constant level to $\mathcal{O}(w)$. Suppose each layer has $w$ neurons: $w_1=w_2=\cdots=w_k=w$, and all neurons in each layer are intra-linked, the upper bound of the intra-layer linked network is $(\frac{(w+1)w}{2}+1)^k$, which approximately equals to that of a feedforward network with width $w_1=w_2=\cdots=w_k=\frac{(w+1)w}{2}$. At this time, the improvement of representation power by intra-links is $\mathcal{O}(w)$ instead of a constant-level improvement. Thus, the separation is still valid if one allows increasing the width of feedforward networks by a constant factor.
>
> - The improvement is exponentially dependent on depth. Even if only every two neurons are intra-linked in a layer, the upper bound of the number of pieces by a network of  $k$  hidden layers with widths  $w_{1}, \ldots, w_{k}$ is $\prod_{i=1}^{k}\left(\frac{3w_{i}}{2}+1\right)$. The improvement is approximately $\mathcal{O}(\frac{3}{2})^k$, which is considerable when a network is deep.
>
> - Please note that adding intra-layer links increases no trainable parameters for a model, which is an economic add-on to the model, and has the potential of enhancing model compactness.
>
> **[Q3] In Section 4.1.3, it is only shown that every standard feedforward ReLU network can also be represented with a network with intra-links and the same depth and width. I would not view this as a valid separation result.**
>
> Sorry for the confusion. Our paper consists of two parts. First, we substantiate that adding intra-layer links can greatly increase the number of pieces via bound estimation, explicit construction, and functional space analysis. Then, adding intra-layer links can empower the shallow networks to represent complicated functions such as sawtooth functions, without the need of going as wide as before. The result you mentioned is a result to show that adding intra-layer links can enlarge the functional space.
>
> **[Q4] Minor question: in the end of remark 1, I was wondering if further elaboration could be made to clarify what exactly “needs to be re-examined”.**
>
> Arora et al. [1] claimed the following theorem:
> For every pair of natural numbers  $k \geq 1, w \geq 2$, there exists a function representable by an  $\mathbb{R} \rightarrow \mathbb{R}$ $(k+1)$-layer feedforward ReLU DNN of width  $w$  such that if it is also representable by a $\left(k^{\prime}+1\right)$-layer feedforward ReLU DNN for any $k^{\prime} \leq k$, this $\left(k^{\prime}+1\right)$-layer feedforward ReLU DNN has width at least  $\frac{1}{2} w^{\frac{k}{k^{\prime}}}$.
>
> Since Arora et al. (2016)’s bound for the number of pieces generated by a standard feedforward ReLU network is loose, the gap in their depth separation result is large. The above lower bound $\frac{1}{2} w^{\frac{k}{k^{\prime}}}$ should be $w^{\frac{k}{k^{\prime}}}$ based on our tight estimation.
>
> [1] Raman Arora, Amitabh Basu, Poorya Mianjy, and Anirbit Mukherjee. Understanding deep neural
> networks with rectified linear units. arXiv preprint arXiv:1611.01491, 2016.

---

> > ### Comment · Reviewer_p5Vt · 2022-11-13
> > **Response**
> >
> > Hi authors,
> >
> > I'm glad to see some new results showing that adding $n$ intra-links can improve the representation power from constant level to $O(n)$ level (Q2 above). This is indeed interesting and improves the previous separation result. I was wondering if authors could give the proof of Proposition 6 in Appendix F. In particular, I wonder if authors could show the tightness of Theorem 15.1 in Appendix F as currently this is only an upper bound on the number of linear pieces. Based on my understanding, only after showing the tightness (i.e., the upper bound could be achieved), one could then use it to show Proposition 6. I'm willing to increase my score if the proof could be provided.

---

> > > ### Author Response · Authors · 2022-11-15
> > > **Thanks for your response. Proposition 6 is proved**
> > >
> > > Dear Reviewer p5Vt:
> > >
> > > We would like to thank you for your quick response and recognition of our new results. Per your suggestion, we have proved the enhanced separation result (Proposition 6) ([Q1]). Now, the power of intra-layer links is more convincing. Please refer to Appendix F for details. Please don't hesitate to let us know if you have additional comments, and we are always happy to discuss them here.
> > >
> > > **[Q1]: I was wondering if the authors could give the proof of Proposition 6 in Appendix F.**
> > >
> > > Yes! In Theorem 16 of Appendix F, we have shown that the bound $\prod_{i=1}^{k}\left(\frac{(w_i+1)w_{i}}{2}+1\right)$ is tight for a one-hidden-layer intra-linked network by offering an explicit construction to generate as many as $\frac{w_1(w_1+1)}{2}+1$ breakpoints, where $w_1$ is the number of neurons. Because a feedforward network with $w_1$ neurons can maximally generate $w_1+1$ pieces, the enhanced depth separation suggested by Proposition 6 is proved. Last but not least, we spotlight that no matter how many neurons are linked, the improvement of depth separation by intra-layer links increases no trainable parameters for a network.
> > >
> > > **[Q2]: Tightness of Theorem 15.1 in Appendix F.**
> > >
> > > Not satisfied with the tightness of Theorem 15.1 for a one-hidden-layer intra-linked network which is already sufficient to prove Proposition 6, we have further proved two results. An arbitrarily deep network of width=3 and with all neurons in each layer intra-linked can achieve $5^k$ pieces (Theorem 18), which is very close to the upper bound for this kind of network ($7^k$). An arbitrarily deep network of width=4 and with all neurons in each layer intra-linked can achieve $9^k$ pieces (Theorem 17), which is also very close to the upper bound for this kind of network ($11^k$). Indeed, by careful parameter calibration, we can generalize the results into $n=5,6,\cdots$ to get approximately $14^k, 20^k, \cdots$ pieces (The upper bounds are $16^k, 22^k, \cdots$). However, for this moment, we cannot find the unified mathematical induction approach to show the tightness of Theorem 15.1 for the arbitrarily wide and deep intra-linked network. We leave this difficult problem as our future exploration.
> > >
> > >
> > > Please kindly let us know if you have any additional questions regarding the new results. Thank you!
> > >
> > > Sincerely,
> > >
> > > Anonymous Authors

---

> > > > ### Comment · Reviewer_p5Vt · 2022-11-16
> > > > **response**
> > > >
> > > > Thanks for the response and providing the proof for Proposition 6. I will increase my score.

---

### Official Review · Reviewer_GyBS · 2022-11-01

**Confidence:** 4
**Correctness:** 3
**Technical Novelty And Significance:** 3
**Empirical Novelty And Significance:** Not applicable
**Recommendation:** 6

**Clarity, Quality, Novelty And Reproducibility:**

Clarity:
This paper is well-written. The definitions, assumptions, and theorems are clearly stated.

Novelty:
As far as I know, this paper is the first to analyze the effects of intra-layer links in representation power.

Reproducibility:
I think the theoretical results are sound and reproducible.

**Strength And Weaknesses:**

Strengths:

This paper is well-written and the theory is sound. It shows that just adding very simple intra-layer links to the neurons (without adding any new parameter) can already significantly increase the representation power of the network. Most previous works on depth separation focus on fully connected neural networks. This work supplements those previous results by considering a network with intra-layer links.

Weaknesses:

1. I cannot agree with the authors' definition of network depth. If we view a neural network as a directed acyclic graph, then I think its depth is just the length of the longest path from the input to the output. With this definition, adding the intra-layer links certainly increases the depth of the network by roughly a factor of two. Therefore, I view the addition of intra-layer links as a particular way of increasing depth, which is still different from stacking more layers in feed-forward networks. From this perspective, I think this paper shows that increasing the network depth in a very simple and restricted manner (intra-layer links) can still increase the network's representation power. This paper mentioned that adding intra-layer links and stacking more layers though both increase the representation power, have a very different mechanism for doing so. Although a short remark was given, it might be better to explain more and maybe add figures to illustrate the different effects between intra-layer links and stacking more layers on the piece-wise linear function.
2. I found the motivation of this paper not very clear. There are many different ways to increase the network depth other than simply stacking more layers, why do we want to analyze the addition of intra-layer links? This kind of intra-layer link is not used in practice.





**Summary Of The Paper:**

1. This paper proves that introducing intra-layer links to the neurons of the same layer allows the network to represent piece-wise linear functions with more pieces, via both upper-bound estimation and explicit construction. Furthermore, by introducing intra-links to the first layer, the new network can represent any function representable by the old network, and more.
2. This paper modified the previous depth separation results by showing that to represent a function constructed by a deep network, a shallow network with intra-layer links only needs 2/3 of the width required for a normal shallow network

**Summary Of The Review:**

This paper shows that adding intra-layer links to a network can increase its representation power. Since most previous works on depth separation have focused on feed-forward networks, this paper supplements the literature by considering a different architecture.

By a standard definition of network depth, adding intra-layer links does increase network depth. Since there are so many different ways to increase network depth other than stacking more layers, the motivation for considering the addition of intra-layer links is unclear to me.

---

> ### Author Response · Authors · 2022-11-08
> **Response to Reviewer GyBS**
>
> Dear Reviewer GyBS:
>
> Thank you for your recognition of the significance, novelty, and soundness of our work. We have addressed all of your concerns in the rebuttal, and also will revise the paper per your comments. For more details, please check out Section E in the Appendix. Please feel free to let us know if you have additional comments, and we are always happy to discuss them here.
>
> **[Q1] I cannot agree with the authors' definition of network depth. If we view a neural network as a directed acyclic graph....Therefore, I view the addition of intra-layer links as a particular way of increasing depth, which is still different from stacking more layers in feed-forward networks.**
>
> We agree with you that topologically, one can define the depth of a network as the length of the longest path from the input to the output, by regarding a neural network as a directed acyclic graph. Thus, adding the intra-layer links certainly increases the depth of the network by roughly a factor of two. However, if we examine the operations along the longest path, the number of depth-defining operations remains unchanged in the context of intra-layer links. In a feedforward network, adding a new layer means that the depth increases by 1, and the characteristic of stacking a new layer is adding the affine transformation followed by activation. A feedforward network with two layers involves two times of (affine transformation, activation). In contrast, adding intra-layer links in a fully-connected layer actually exerts a gating effect. When $\sigma(W_2x+b_2)>0$, the output is $\sigma((W_1+W_2)x+b_1+b_2)$; when $\sigma(W_2x+b_2)=0$, the output is $\sigma(W_1x+b_1)$. Essentially, the number of (affine transformation, activation) is still one.
> Thus, if the depth is defined as the number of (affine transformation, activation) that are actually executed, the depth of the intra-linked network is the same as that of the feedforward network. Since topologically one can make any shallow network arbitrarily deep by making a series of reducible identity layers, we think that the intrinsic computational operation is more important than the extrinsic topology for the depth separation theory.
>
> **[Q2] This paper mentioned that adding intra-layer links and stacking more layers though both increase the representation power, have a very different mechanism for doing so. Although a short remark was given, it might be better to explain more and maybe add figures to illustrate the different effects between intra-layer links and stacking more layers on the piece-wise linear function.**
>
> Thanks for this wonderful comment that inspires us to consider the essential differences between stacking a layer and inserting intra-layer links. We think that stacking a layer and inserting intra-layer links are essentially different in the following aspects:
>
> - Their mechanisms of producing pieces are fundamentally different. While the mechanism of adding a new layer is the repetition effect (multiplication), i.e., the function value of the function being composed is oscillating, and each oscillation can generate more pieces. The mechanism of intra-layer links is the gating effect (addition). The neuron being embedded have two activation states, and each state is leveraged by the neuron being linked to produce a breakpoint. Two states are integrated to generate more pieces.
>
> - The function classes (the set of functions represented by some given neural network architecture) of our intra-linked network and the deeper feedforward network are not the same, and this will make a big difference. In some sense, the deeper feedforward network has a larger function class, and the function class of our intra-linked network is just a subset of it. However, our intra-linked network has more expressive power (i.e., number of pieces, VC dimension) per parameter. Also, the experiments (Section D of the Appendix) showed that intra-linked networks can achieve better accuracy in solving real-world problems. This phenomenon can be seen as an analog to the comparison between CNNs and fully-connected NNs. The function classes of CNNs are just subsets of the function classes of fully-connected NNs with some further restrictions on the weights. However, CNNs usually have more expressive power per parameter and achieve better results in practice.
>
> **[Q3] I found the motivation of this paper not very clear. There are many different ways to increase the network depth other than simply stacking more layers, why do we want to analyze the addition of intra-layer links?**
>
> Sorry for the confusion. Our work is motivated by the topology of ResNet. We think that intra-layer links are an extension of residual connections. The former is to link the neurons inside a layer, while the latter is to connect neurons across layers. Also, we take the residual connections in ResNet as outer links and the intra-layer links as inner links.

---

> > ### Comment · Reviewer_GyBS · 2022-11-16
> > **Thank you for the response**
> >
> > Thank you for the response! I agree there are different ways to define network depth. And thank you for explaining the different mechanisms between stacking layers and inserting intra-layer links to increase the representation power. But I think the motivation for studying intra-layer links is still not convincing. The authors mentioned ResNet, but ResNet added outer links without increasing the network depth (defined as the longest path from input to output). Therefore, I think I will keep the current score.

---

> > > ### Author Response · Authors · 2022-11-17
> > > **Clarification of our motivation**
> > >
> > > Dear Reviewer GyBS:
> > >
> > > Thanks for your comments! Now let us illustrate in detail how our work is motivated and what role the ResNet plays in our motivation:
> > >
> > >
> > > - **(Rethinking depth separation theory)** Depth separation theories are widely-accepted theories to address the superiority of depth over width. In many successful network architectures, shortcut connections are used. However, few studies, if not none, considered depth separation theories in the shortcut paradigm. Our curiosity is will the depth separation still hold when shortcuts are used? Along this direction, we find that the residual connections in ResNet are outer links embedded into a network horizontally. The shallow network with residual connections can have comparable performance with a deep network, i.e., the insertion of residual connections can save depth. Based on a consideration of structural symmetry, we embed shortcuts vertically, i.e., intra-linking neurons within a layer. Our work reveals that topologically, the insertion of intra-layer links can save width. This is why we can lower the threshold and modify the depth separation theory.
> > >
> > > - **(Exploring new architectures)** Exploring new and powerful network architectures such as the invention of ResNet has been the mainstream research direction in deep learning in the past decade. Although shortcuts have been widely adopted in network design, to the best of our knowledge, we are the first to consider adding shortcuts within a layer in a fully-connected network. Like the residual connections, we spotlight that no matter how many neurons are linked, the improvement of representation power by intra-layer links increases no trainable parameters for a network. Thus, the intra-layer link is an extremely economical add-on to the model, which has the great potential of enhancing model compactness. Furthermore, our new results (Appendix F of the rebuttal revision) demonstrate that if intra-linking more neurons in a layer, the improvement can go from the constant level to $\mathcal{O}(w)$, where $w$ is the width. Finally, our added experiments (Appendix D of the rebuttal revision) show that the intra-linked network can indeed improve the model's prediction accuracy in solving a real-world task.
> > >
> > > - **(Explaining the power of shortcuts)** Well-established network architectures such as ResNet and DenseNet imply that incorporating shortcuts greatly empowers a neural network in solving real-world problems. However, theoretical studies are few to explain the representation ability of shortcuts. In this study, we would like to fill this gap. We investigate what will happen after inserting shortcuts within a layer, thereby shedding light on the mechanism and characteristics of shortcuts. As one of our research outcomes, the mechanism of generating more pieces we identified can be translated into other shortcut networks such as ResNet (Appendix G of the rebuttal revision).

---

### Official Review · Reviewer_EkgR · 2022-11-01

**Confidence:** 4
**Correctness:** 2
**Technical Novelty And Significance:** 2
**Empirical Novelty And Significance:** Not applicable
**Recommendation:** 3

**Clarity, Quality, Novelty And Reproducibility:**

- I haven't checked the proofs in detail, but the paper is quite clear.
- As far as I can tell the result is novel extension of previously known theorems, albeit this novelty is likely due to the type of architectures being investigated not being useful or interesting.
- Reproducibility doesn't apply since this is a theoretical paper.

**Strength And Weaknesses:**

Strengths:
- the paper is clear

Weakness:
- Contrary to what is claimed in the paper, I think that the architecture being presented is essentially equivalent to a standard ReLU DNN with twice depth and weight matrices in a specific form.
- I fail to see the relevance of this work. It focuses on an unusual network architecture which, as far as I can tell, is not used in practice and seems to have been invented by the authors. There are no obvious benefits in using such architecture, in fact it would likely result in higher implementation complexity and/or a performance loss. The theoretical improvement is small and probably not worth just increasing the depth of a standard implementation of a ReLU DNN.


**Summary Of The Paper:**

In this paper the authors theoretically investigate the representation function of a class of neural networks, which as far as I know is novel, that they call intra-linked ReLU DNN.
They define intra-linked ReLU DNN as a modification of a standard feed-foward ReLU multi-layer perceptron, and they prove variations of the separation theorems of Telgarsky (2015) and Arora et al. (2016) that relate width and depth, showing that their intra-linked ReLU DNN has a small constant multiplicative improvement over a standard ReLU DNN.


**Summary Of The Review:**

The paper is an extension of existing theoretical results to a seemingly uninteresting architecture.

---

> ### Author Response · Authors · 2022-11-08
> **Response to Reviewer EkgR**
>
> Dear Reviewer EkgR:
>
> We are sorry for some misunderstandings in the current version of the paper. We have addressed all of your concerns in the rebuttal, and also will revise the paper following your comments. For more details, please check out Sections D and E in the Appendix. We are looking forward to discussing this draft with you further.
>
> **[Q1] The architecture being presented is essentially equivalent to a standard ReLU DNN with twice depth and weight matrices in a specific form.**
>
> We agree with you that a network with intra-layer links can be unfolded into a dense network with a special arrangement. However, we find that inserting intra-layer links is intrinsically different from stacking a new layer in the following aspects:
>
>  - Their mechanisms of producing pieces are fundamentally different. While the mechanism of adding a new layer is the repetition effect (multiplication), i.e., the function value of the function being composed is oscillating, and each oscillation can generate more pieces. The mechanism of intra-layer links is the gating effect (addition). The neuron being embedded have two activation states, and each state is leveraged by the neuron being linked to produce a breakpoint. Two states are integrated to generate more pieces.
>
> - Although both involve composition, they use different numbers of (affine transform, activation). In a feedforward network, stacking a new layer means the depth increases by 1, and the characteristic of stacking a new layer is doing the affine transformation followed by activation. A feedforward network with two layers involves two times of (affine transformation, activation). In contrast, adding intra-layer links in a fully-connected layer actually exerts a gating effect. When $\sigma(W_2x+b_2)>0$, the output is $\sigma((W_1+W_2)x+b_1+b_2)$; when $\sigma(W_2x+b_2)=0$, the output is $\sigma(W_1x+b_1)$. The number of (affine transformation, activation) is still one for both cases.
>
> - The function classes (the set of functions represented by some given neural network architecture) of our intra-linked network and the deeper feedforward network are not the same, and this will make a big difference. In some sense, the deeper feedforward network has a larger function class, and the function class of our intra-linked network is just a subset of it. However, our intra-linked network has more expressive power (i.e., number of pieces, VC dimension) per parameter. This phenomenon can be seen as an analog to the comparison between CNNs and fully-connected NNs. The function classes of CNNs are just subsets of the function classes of fully-connected NNs with some further restrictions on the weights. However, CNNs usually have more expressive power per parameter and achieve better results in practice.
>
> Topologically, one can define the depth of a network as the length of the longest path from the input to the output, by regarding a neural network as a directed acyclic graph. Thus, adding the intra-layer links certainly increases the depth by roughly a factor of two. However, if we examine the operations along the longest path, the number of (affine transformation, activation) remains to be the same with the number of layers. Thus, if the depth is defined as the actual number of (affine transformation, activation), the depth of the intra-linked network is the same as that of the feedforward network. Since topologically a network can be made arbitrarily deep by making a series of reducible identity layers, we argue that the intrinsic computational operation is more important than the extrinsic topology for the depth separation theory.
>
> **[Q2] I fail to see the relevance of this work. It focuses on an unusual network architecture which, as far as I can tell, is not used in practice and seems to have been invented by the authors. There are no obvious benefits in using such architecture, in fact it would likely result in higher implementation complexity and/or a performance loss.**
>
> - The intra-layer link is not exactly an unusual architecture. It can be regarded as an extension of residual connections. The classical residual connection is to add residual links across layers, while intra-layer links are to add residual links with a layer. The residual connections in ResNet are outer links and the intra-layer links serve as inner links.
>
> - We have implemented the intra-linked network and compared it with a feedforward network in a real-world task. Please refer to our **Response to Reviewer 4Twm [Q1]**. The implementation of intra-layer links is pretty straightforward, since intra-layer links are a simple add-on to a network. No performance loss shows up in our experiments.
>
> **[Q3] The theoretical improvement is small and probably not worth just increasing the depth of a standard implementation of a ReLU DNN.**
>
> To avoid repetition, please refer to our **response to Reviewer p5Vt [Q2]**.

---

> > ### Author Response · Authors · 2022-11-15
> > **New results on the theoretical improvement of intra-layer links**
> >
> > Dear Reviewer EkgR,
> >
> > We are very grateful for your constructive feedback! We have generated new results to address your concern about the small theoretical improvement by intra-layer links. Please check out Appendix F for details. We look forward to discussing with you.
> >
> > **[Q3] The theoretical improvement is small and probably not worth just increasing the depth of a standard implementation of a ReLU DNN.**
> >
> > - Adding more intra-layer links can boost the gain from the constant level ($\frac{3}{2}$) to $\mathcal{O}(w)$. Suppose each layer has $w$ neurons: $w_1=w_2=\cdots=w_k=w$, and all neurons in each layer are intra-linked, the upper bound of the intra-layer linked network is $(\frac{(w+1)w}{2}+1)^k$ (Theorem 15 of Appendix F), while the upper bound of a feedforward network with width $w$ is $(w+1)^k$. We verify the tightness of this upper bound for the one-hidden-layer intra-linked network (Theorem 16), a network of width 3 and arbitrary depth (Theorem 18), and a network of width 4 and arbitrary depth (Theorem 17). Thus, the intra-linked ReLU DNN has an $\mathcal{O}(w)$ instead of a small constant multiplicative improvement over a standard ReLU DNN, which reveals the genuine power of intra-layer links.  Thanks to the $\mathcal{O}(w)$-level improvement, intra-linking all neurons with a layer is approximately equal to doubling the network depth.
> >
> > - Please note that no matter how many neurons are intra-linked, adding intra-layer links increases no trainable parameters for a model, which is an extremely economic add-on to the existing model.  In contrast, stacking layers comes with the cost of increasing parameters. We highlight that inserting intra-layer links has more parameter efficiency than stacking layers.
> >
> > - The improvement is exponentially dependent on depth. Even if only every two neurons are intra-linked in a layer, the upper bound of the number of pieces by a network of  $k$  hidden layers with widths  $w_{1}, \ldots, w_{k}$ is $\prod_{i=1}^{k}\left(\frac{3w_{i}}{2}+1\right)$. The improvement is approximately $\mathcal{O}(1.5^k)$, which is a considerable gain when a network is deep.
> >
> > Sincerely,
> >
> > Anonymous Authors

---

> ### Author Response · Authors · 2022-11-16
> **The Second Response to Reviewer EkgR**
>
> Dear Reviewer EkgR:
>
> We would like to thank you for your comments. Here, we address your concern about the novelty of our work, and also will revise the paper following your comments. Please re-evaluate our draft based on our clarification and new results. If you have any further additional suggestions, we are happy to discuss them here.
>
>
>
> **[Q4] They prove variations of the separation theorems of Telgarsky (2015) and Arora et al. (2016) that relate width and depth; The result is novel extension of previously known theorems.**
>
> Our work is not just a novel extension of the previously known depth separation theorems. It also makes unique conceptual and technical contributions as the following:
>
> - To the best of our knowledge, our work is the first to consider the depth separation in the context of shortcut paradigm, which non-trivially supplements the depth separation theory. In reality, a neural network often is not feedforward but uses shortcuts to link distant layers to facilitate feature reuse and easy training. Exploring the depth separation in the shortcut paradigm reveals the limit of the existing theory and motivates practitioners to rethink the genuine power of depth.
>
> - Our results are established by applying novel analysis techniques which can be translated into analyzing networks of other architectures such as ResNet. First, in every activation and pre-activation step, we carefully distinguish the repeated breakpoints and estimate the maximum number of newly generated breakpoints, thus offering a much tighter bound than those in previous studies. Second, following the idea of bound estimation, we systematically design procedures to produce most pieces in the activation step while preserving all the existing breakpoints in the affine combination step. The resultant construction either reaches the upper bound or is at the same magnitude. Third, we analyze the mechanism and functional space of networks with or without links to illustrate the representation ability of intra-layer links. In Appendix G, these techniques are applied to analyze the ResNet and show that residual connections can greatly boost the representation power of a network.

---

### Official Review · Reviewer_4Twm · 2022-11-04

**Confidence:** 1
**Correctness:** 4
**Technical Novelty And Significance:** 3
**Empirical Novelty And Significance:** Not applicable
**Recommendation:** 6

**Clarity, Quality, Novelty And Reproducibility:**

I can't adequately judge the clarity of the mathematical theorems and proofs, but the major findings and description of the proposed architecture could be introduced a bit more clearly. I believe the work is quite novel.

**Strength And Weaknesses:**

Strengths:
- Proposes novel theoretical architecture and does significant theoretical analysis of it's representative power in relation to shallow and deep networks


Weaknesses:
- This theoretical architecture would not be difficult to implement, so the work may benefit from a real implementation and comparison with dense networks on at least a toy example.


**Summary Of The Paper:**

This work proposes a theoretical modification of the dense feedforward network whereby each neuron in each layer in a densely connect feed forward network is paired with one other Neurons. The output of one Neuron in each pair is overwritten by the sum of the 2 linear outputs of both Neurons before being passed to the activation function. They demonstrate that for the same depth, intra-linked networks have the linear regions it could possibly represent is high. For $n$ dimensional input, $w$ widths, and $k+1$ layers, they derive a upper bound of $\prod_{i=1}^k{\sum_{j=0}^n\left(\frac{\frac{3w_i}{3} + 1} {j}\right)}$ for intra-linked vs a previous derived upper bound of $\prod_{i=1}^k{\sum_{j=0}^n\left(\frac{w_i} {j}\right)}$ for dense networks. They derive that for networks of width 2 and $k+1$ layers, a regular feed forward network can represent a sawtooth function with at most $\sqrt{6}^k$ if k is even or $3 \cdot \sqrt{7}^{k-1}$ pieces if k is odd. They show that furthermore, a intra-linked network of the same dimensions can produce a sawtooth function of at least $7 \cdot 3^{k-2}$ pieces. They derive a proof for all $k\ge1$, their exists a function represented by a classical network of width 6, and $k^2 +1$ layers that can't be represented by a classical $k+1$ network of width less than $6^k$  that can be represented by some intra-linked network of width less than $4 \cdot 6^{k-1}$. They derive a proof for all $k\ge2$, their exists a function represented by a classical network of width 2, and $k^2 +1$ layers that can't be represented by a classical 3 layer network of width less than $k-1$  that can be represented by some intra-linked network of 2 layers with width of $\frac{2(k-1)}{3}$.

**Summary Of The Review:**

This work proposes novel theoretical architecture and does significant theoretical analysis of it's representative power in relation to shallow and deep networks. Unfortunately, I don't believe I can very accurately judge the importance and accuracy of the mathematical theorems and proofs introduced in this paper. The work tackles an important area of furthering our theoretical understanding the expressive power and effectiveness of deep networks and suggest towards further research in the domain of shallow networks. The architecture is not very complicated however, and seems like a basic experimental demonstration should be quite feasible, but is currently not provided.

---

> ### Author Response · Authors · 2022-11-08
> **Response to Reviewer 4Twm**
>
> Dear Reviewer 4Twm:
>
> We would like to thank you for your recognition of the novelty and significance of our work. Our result non-trivially supplements the depth separation theory in the setting of shortcuts. Also, many thanks for your constructive suggestion of adding an example for comparison between intra-linked and feedforward networks. We have implemented two networks to solve a real-world task and will include such an example in the paper following your suggestion:
>
>
> **[Q1]This theoretical architecture would not be difficult to implement, so the work may benefit from a real implementation and comparison with dense networks on at least a toy example.**
>
> Thanks for your suggestion. We validate whether or not the intra-layer links can assist a network to deliver superior performance in real-world tasks. The task is to predict if a credit card holder will get churned so that the bank can provide better service to turn holders' decisions. This prediction task has 10,000 raw samples, and each has 18 customers' portfolio features including age, salary, marital status, credit card limit, credit card category, etc. The labels are 'get churned' or 'stay'. The detailed description of data and this task can be referred to in Kaggle (https://www.kaggle.com/datasets/whenamancodes/credit-card-customers-prediction).
>
> The data are preprocessed as follows: The discrete value is assigned to different education levels based on the mapping \{
>     'Uneducated': 0,
>     'High School': 1,
>     'College': 2,
>     'Graduate': 3,
>     'Post-Graduate': 4,
>     'Doctorate': 5
> \}. The income situation is assigned with values based on the mapping: \{
>     'Less than 40K ': 0,
>     '40K - 60K ': 1,
>     '60K - 80K ': 2,
>     '80K - 120K ': 3,
>     '120K +': 4
> \}. The female and male are mapped to 0 and 1, respectively. All samples with missing attributions are deleted. Finally, the processed data have 7,081 data points.  Then, the data are randomly split into training and testing sets with a ratio of 0.8:0.2.
>
> We build networks with intra-layer links and compare them with the corresponding feedforward networks without intra-layer links. The optimizer is Adam with a learning rate of 0.1. The loss function is the binary cross-entropy function. The number of epochs is 1500 to ensure convergence. The evaluation metric is the widely used micro-F1 score.
>
> The prediction results by feedforward and intra-linked networks are summarized in the table below, from which we draw two highlights. First, when the same structure is used, the intra-linked network consistently outperforms the feedforward network. More favorably, the intra-linked network takes the lead by a large margin: the minimum gain is 1.68\% for the network structure 22-32-2, while the maximum gain is 2.66\% for the network structure 22-16-16-2. Such a superiority corroborates our theoretical analysis that adding intra-layer links can boost the network's representation power. Second, the one-hidden-layer intra-linked network can sometimes perform superbly over the two-hidden-layer feedforward network (22-16-2 vs 22-8-8-2, 22-32-2 vs 22-16-16-2, 22-64-2 vs 2-232-32-2), when their parameters are comparable. This phenomenon suggests that the representation power of an intra-linked network is different from that of a feedforward network with more layers.
>
>
> | Feedforward   |      #Parameters      | Micro-F1(%) | Intra-linked    |    #Parameters     |  Micro-F1(%) |
> |----------|-------------|----------------------|-------------------|-------------|------------------|
>   |22-8-2 | 202| 78.24$\pm$0.23\% | 22-8-2 | 202| **80.13$\pm$0.37\%**
>  | 22-16-2   | 402| 78.93$\pm$0.19\% |22-16-2 |  402| **80.83$\pm$0.41\%**
>    | 22-32-2   | 802| 80.53$\pm$0.18\% | 22-32-2 |  802| **82.21$\pm$0.27\%**
>   |22-64-2 | 1602| 82.03$\pm$0.23\% | 22-64-2 | 1602| **83.52$\pm$0.34\%**
>   |22-8-8-2 | 274|  80.21$\pm$0.32\%  | 22-8-8-2 | 274| **82.11$\pm$0.42\%**
>   |22-16-16-2 | 674| 81.14$\pm$0.34\%  | 22-16-16-2 | 674| **83.80$\pm$0.37\%**
>    | 22-32-32-2 | 1858 | 83.14$\pm$0.28\%  | 22-32-32-2 | 1858 |**84.79$\pm$0.45\%**
>    | 22-64-64-2 |5762 | 84.35$\pm$0.24\%  | 22-64-64-2 | 5762 |**85.44$\pm$0.25\%**

---

### Author Response · Authors · 2022-11-15
**General Response (To Reviewer 4Twm&EkgR&GyBS&p5Vt)**

Dear Reviewers:

Thanks for recognizing the novelty and contributions of our draft and making so many helpful comments that motivate us to further enhance our draft.  Our work is the first to consider depth separation theory in the shortcut regime. It shows that just adding very simple intra-layer links to the neurons (without adding any new parameter) can already significantly increase the representation power of the network, therefore, a shallow network does not need to go as wide as before to have a representation power on a par with a deep network. We have generated new results to address all your concerns and substantially improve the quality of our draft. Please check out Appendices D, E, F, and G for details. We will revise our paper accordingly based on your comments. We believe a reevaluation of our paper is highly necessary. We look forward to discussing this with you!

**[Reviewer 4Twm] The usefulness of the intra-linked networks**

We have validated whether or not the intra-layer links can assist a network to deliver superior performance in real-world tasks (Appendix D). The results show that adding intra-layer links can boost the network's predictive performance by a large margin.

**[Reviewer EkgR&GyBS] The differences between stacking layers and inserting intra-layer links**

While a network with intra-layer links can be unfolded into a dense network with a special arrangement,  inserting intra-layer links is intrinsically different from stacking a new layer in terms of the fundamental mechanism of generating new pieces, the number of affine transforms being used, and the functional class (Appendix E).

**[Reviewer GyBS&p5Vt]  The improvement of intra-layer links**

We have shown that adding more intra-layer links can boost the gain from the constant level to $\mathcal{O}(w)$, where $w$ is the number of neurons (Appendix F). Based on this new result, the gap between the intra-linked network and feedforward network in our depth separation results is greatly enlarged from a constant-level saving to $\mathcal{O}(w)$. Furthermore, no matter how many neurons are intra-linked, adding intra-layer links increases no trainable parameters for a model, enjoying more parameter efficiency than stacking layers.

---

### Decision · Program_Chairs · 2023-01-20

**Decision:**

Reject

**Justification For Why Not Higher Score:**

The theoretical results are not particularly strong - the improvements are only in terms of constant factors in width.

**Justification For Why Not Lower Score:**

N/A

**Metareview: Summary, Strengths And Weaknesses:**

This paper considers neural networks with intra-layer connections and how they change the results of depth separation. The main result of the paper shows that for a network with some intra-layer connections (in particular, for pairs of nodes in a layer, connect one to another), even though the number of parameters is not larger, it can represent some functions (used in depth separation constructions) with fewer layers/neurons compared to standard fully connected neural networks. After the author response period, the authors added some results extending intra-layer connections to a more general setting, some basic results for resnet (with 1 neuron per layer) and some empirical justifications for the architecture. The reviewers agree that there are some new ideas in the architecture and results. However, the improvement in depth/number of neurons is not very significant especially in the original setting (for example the gaps in Theorem 12/13 are not more than a constant factor). It would be interesting to see if there is possibility for a larger gap, or if there can be results for resnet which is widely used.

**Summary Of Ac-Reviewer Meeting:**

Reviewer EkgR did not respond to my messages/emails so he/she was not in the reviewer meeting. All the other reviewers attended and their opinions are quite similar. We went through the updated results together and decided that they did not fully address the concern of reviewer EkgR.